# DyG-Mamba: Continuous State Space Modeling on Dynamic Graphs

**Dongyuan Li**[1], **Shiyin Tan**[2], **Ying Zhang**[3], **Ming Jin**[4],
**Shirui Pan**[4], **Manabu Okumura**[2], **Renhe Jiang**[1*]

[1]The University of Tokyo,   [2]Institute of Science Tokyo,
[3]RIKEN Center for Advanced Intelligence Project,   [4]Griffith University

lidy@csis.u-tokyo.ac.jp,  tanshiyin@lr.pi.titech.ac.jp, ying.zhang@riken.jp,
mingjinedu@gmail.com,  s.pan@griffith.edu.au,  oku@pi.titech.ac.jp,
jiangrh@csis.u-tokyo.ac.jp

## Abstract

Dynamic graph modeling aims to uncover evolutionary patterns in real-world systems, enabling accurate social recommendation and early detection of cancer cells. Inspired by the success of recent state space models in efficiently capturing long-term dependencies, we propose DyG-Mamba by translating dynamic graph modeling into a long-term sequence modeling problem. Specifically, inspired by Ebbinghaus' forgetting curve, we treat the irregular timespans between events as control signals, allowing DyG-Mamba to dynamically adjust the forgetting of historical information. This mechanism ensures effective usage of irregular timespans, thereby improving both model effectiveness and inductive capability. In addition, inspired by Ebbinghaus' review cycle, we redefine core parameters to ensure that DyG-Mamba selectively reviews historical information and filters out noisy inputs, further enhancing the model's robustness. Through exhaustive experiments on 12 datasets covering dynamic link prediction and node classification tasks, we show that DyG-Mamba achieves state-of-the-art performance on most datasets, while demonstrating significantly improved computational and memory efficiency. Code is available at [https://github.com/Clearloveyuan/DyG-Mamba].

## 1 Introduction

Dynamic graph modeling represents entities as nodes and timestamped relationships as edges, aiming to explore the underlying evolution patterns of real-world systems [1]. It has attracted great attention in various fields, *e.g.*, social networks [2], traffic systems [3], and recommender systems [4].

Despite the great success of current methods, there are still two limitations. ***Firstly, existing methods lack the ability to effectively and efficiently track long-term temporal dependencies in dynamic graphs.*** Specifically, RNN-based methods, *e.g.*, JODIE [2] and TGN [5], model temporal evolution through recurrent updates of node embeddings. Although theoretically capable of capturing long-term dependencies, they suffer from vanishing/exploding gradients in practice, limiting their effectiveness on long sequences. On the other hand, Transformer-based models, *e.g.*, DyGFormer [6] and SimpleDyG [7], address gradient issues through the self-attention mechanism but require prohibitive quadratic $\mathcal{O}(N^2)$ computational complexity for sequences of length $N$. Recent efficiency improvements through patching [6] or temporal convolutions [8] inevitably sacrifice temporal resolution, forcing an effectiveness-efficiency trade-off. Other recent methods, using multi-layer perceptions (MLP), *e.g.*, GraphMixer [9], FreeDyG [10], or graph neural networks (GNN), *e.g.*, TGAT [11], primarily focus on short-term dependencies, and their performance often decreases as the sequence

---

[*]Corresponding author.

39th Conference on Neural Information Processing Systems (NeurIPS 2025).

length increases [12]. ***Secondly, existing methods lack robustness against noise***. Real-world dynamic graphs frequently contain various types of noisy events [13]. RNN-based and GNN-based methods are naturally susceptible to noise interference, leading to unstable performance [14, 15]. Although Transformers partially mitigate historical noise via self-attention, they remain susceptible to noisy data and cannot fully eliminate its impact [16]. How to filter out noisy history information more effectively and efficiently remains a challenge [17].

To address these issues, we propose DyG-Mamba, a novel timespan-informed continuous state space model (SSM), for dynamic graph modeling. Firstly, compared to Transformer-based methods that rely on large number of trainable parameters, DyG-Mamba employs only one trainable step size parameter $\Delta t$ to capture forgetting laws of historical information, along with a small set of parameters in the encoder and decoder layers. Under the same GPU memory constraints, DyG-Mamba can directly process the entire long-term sequence without pooling, thereby preserving temporal details and effectively modeling long-term dependencies. Furthermore, inspired by Ebbinghaus' forgetting curve [18], which posits that forgetting is primarily correlated with timespans rather than content, we aim to equip DyG-Mamba with the same forgetting mechanism. Specifically, we design a monotonically increasing and learnable timespan function to redefine $\Delta t$, enabling the dynamic system to automatically learn how to compress historical information across different timespans, *i.e.*, the model forgets historical information in a "fast-then-slow" pattern as the timespan increases, thereby enhancing both its effectiveness and inductiveness. Additionally, compared to Transformer's quadratic time complexity, DyG-Mamba adopts the same hardware-aware parallel scan optimization as Mamba [19], enabling it to efficiently capture long-term dependencies with linear time complexity. Secondly, inspired by Ebbinghaus' review cycle that periodic review can counteract forgetting [20], to further enhance robustness, we redefine SSM's core parameters $B$ and $C$ to be input-dependent and add spectral norm constraints to ensure Lipschitz continuity. This strategy enables DyG-Mamba to selectively review historical information and thus remain robust against noise. Main contributions:

- To the best of our knowledge, we are the first to introduce SSMs for continuous-time dynamic graph modeling. By redefining the core SSM parameters, DyG-Mamba achieves high efficiency and effectiveness in capturing long-term temporal dependencies.

- Inspired by both the forgetting curve and the review cycle that counters it, we propose a timespan-informed continuous SSM that adopts timespans to control system forgetting while incorporating input-dependent parameterization. This design improves DyG-Mamba's capability to model long-term sequences with irregular timespans, enhancing its effectiveness, inductiveness and robustness.

- Extensive experiments on 12 benchmarks show that DyG-Mamba achieves state-of-the-art performance with superior effectiveness and robustness.

Table 1: Comparison of continuous-time dynamic graph baselines from six aspects. With a batch size of 200 and a sequence length of 512, a model is considered time and memory efficient if the running time and memory usage are less than GraphMixer, *i.e.*, running time 250 seconds and memory usage 30,000 MB. Adding 50% noisy temporal edges, the performance drop $< 10\%$ indicates robustness.

| | JODIE | DyRep | TGN | CAWN | TGAT | EdgeBank | GraphMixer | TCL | DyGFormer | DyG-Mamba |
|---|---|---|---|---|---|---|---|---|---|---|
| Long-Term Capability | ✘ | ✘ | ✘ | ✘ | ✘ | ✘ | ✘ | ✘ | ✔ | ✔ |
| Time Efficient | ✔ | ✘ | ✘ | ✘ | ✘ | ✔ | ✔ | ✔ | ✘ | ✔ |
| Memory Efficient | ✔ | ✘ | ✘ | ✘ | ✘ | ✔ | ✔ | ✘ | ✘ | ✔ |
| Irregular timespan Supportive | ✘ | ✘ | ✘ | ✘ | ✘ | ✘ | ✘ | ✘ | ✘ | ✔ |
| Inductive Supportive | ✔ | ✔ | ✔ | ✔ | ✔ | ✘ | ✔ | ✔ | ✔ | ✔ |
| Noise Robust | ✘ | ✘ | ✔ | ✘ | ✘ | ✔ | ✘ | ✘ | ✔ | ✔ |

## 2 Related Work

**Dynamic Graph Modeling**. Discrete-time methods segment the dynamic graph into snapshots at a predetermined time granularity, then employ a GNN (snapshot encoder) with a recurrent module (dynamic tracker) to learn node embedding [21–24]. However, fixing the time granularity in advance ignores the fine-grained temporal order within each snapshot. In contrast, continuous-time methods directly use timestamps to learn node embedding. Based on their neural architectures, they can be categorized into four types, including RNN-based methods, *e.g.*, JODIE [2], GNN-based methods,

*e.g.*, DySAT [22], MLP-based methods, *e.g.*, FreeDyG [10], and Transformer-based methods, *e.g.*, SimpleDyG [7]. Additional techniques, such as ordinary differential equations [25, 26], random walks [27], and temporal point processes [28], have also been introduced to capture continuous temporal information. ***Table 1 provides a detailed comparison between DyG-Mamba and SOTAs***, including JODIE [2], DyRep [29], TGN [5], CAWN [11], TGAT [11], EdgeBank [30], GraphMixer [9], TCL [23], and DyGFormer [6] from the following angles: if the method can effectively handle unseen nodes during training (*i.e.*, inductive), capture long-term dependencies with both time and memory efficiency, exhibit robustness against noise, and effectively leverage irregular timespans.

**State Space Models**. SSMs have attracted great attention for long sequence modeling [31, 32]. Mamba [19] designs a data-dependent selection mechanism with parallel scan optimization, achieving SOTA performance on many fields [33]. Graph Mamba [34, 35] applies SSMs to static graphs for embedding learning. DG-Mamba [17] and GraphSSM [36] extend Mamba to discrete-time dynamic graphs by modeling snapshot sequences with fixed time intervals. And STG-Mamba [37] adopts Mamba layers on spatial-temporal graphs. However, these methods are not applicable to continuous-time dynamic graphs with irregular timestamps, and thus are not directly comparable to our setting. PIVEM [38] learns dynamic node embeddings by approximating temporal evolution through piecewise linear interpolation, based on a latent distance model with piecewise constant and node-specific velocities. It can be viewed as a special case of a first-order SSM, where the hidden state corresponds to node velocity and evolves linearly over time. In contrast, our method generalizes this idea by introducing learnable memory decay and input-adaptive updates, allowing it to better capture irregular temporal dynamics and model more complex patterns in dynamic graphs.

## 3 Preliminary

**Dynamic Graph Modeling.** Dynamic graphs can be modeled as a sequence of non-decreasing chronological interactions $\mathcal{G} = \{(u_1, v_1, t_1), \ldots, (u_\tau, v_\tau, \tau)\}$ with $0 \le t_1 \le \tau$, where $u_i, v_i \in \mathcal{V}$ denote the source and destination nodes of the $i$-th link and $\mathcal{V}$ denote all nodes. Each node is associated with a node feature $\boldsymbol{x} \in \mathbb{R}^{d_N}$ and each interaction has a link feature $\boldsymbol{e}^t \in \mathbb{R}^{d_E}$, where $d_N$ and $d_E$ denote dimensionality. Given the source node $u$, destination node $v$, timestamp $t$, and all their historical interactions before $t$, *dynamic graph modeling* aims to learn time-aware node embedding for them. We validate the learned node embedding via two common tasks: (i) *dynamic link prediction*, which predicts whether two nodes are connected in future; and (ii) *dynamic node classification*, which infers the class of nodes.

**Continuous SSMs.** They define a linear mapping from $t$-th input $\boldsymbol{u}(t) \in \mathbb{R}^{1 \times d}$ to output $\boldsymbol{y}(t) \in \mathbb{R}^d$ via a hidden state variable $\boldsymbol{h}(t) \in \mathbb{R}^{m \times d}$, formulated by:

$$\boldsymbol{h}'(t) = \boldsymbol{A}\boldsymbol{h}(t) + \boldsymbol{B}\boldsymbol{u}(t), \tag{1}$$
$$\boldsymbol{y}(t) = \boldsymbol{C}\boldsymbol{h}(t) + D\boldsymbol{u}(t), \tag{2}$$

where $\boldsymbol{A} \in \mathbb{R}^{m \times m}$, $\boldsymbol{B} \in \mathbb{R}^{m \times 1}$, $\boldsymbol{C} \in \mathbb{R}^{1 \times m}$ are trainable parameters, and $D = 0$ since $D\boldsymbol{u}(t)$ can be viewed as a skip connection. Eq.(1,2) could be discretized for controllable optimization via the zero-order hold (ZOH), formulated by:

$$\boldsymbol{h}_t = \overline{\boldsymbol{A}}\boldsymbol{h}_{t-1} + \overline{\boldsymbol{B}}\boldsymbol{u}_t, \tag{3}$$
$$\boldsymbol{y}_t = \overline{\boldsymbol{C}}\boldsymbol{h}_t, \tag{4}$$

where $\overline{\boldsymbol{A}} = \exp(\boldsymbol{\Delta t A})$, $\overline{\boldsymbol{B}} = (\boldsymbol{\Delta t A})^{-1}(\overline{\boldsymbol{A}} - \boldsymbol{I})(\boldsymbol{\Delta t B})$, $\overline{\boldsymbol{C}} = C$, and $\boldsymbol{\Delta t}$ is predefined step size.

## 4 Methodology

The overview of DyG-Mamba is shown in Figure 1. First, in Section 4.1, we introduce dynamic graph encoding and encoding alignment. Then, in Section 4.2, we introduce two main limitations of current SSMs, and DyG-Mamba can alleviate these issues by redefining four core parameters of SSMs. Finally, in Section 4.3, we apply DyG-Mamba on downstream tasks and show its complexity.

### 4.1 Dynamic Graph Encoding

In Figure 1, we first extract the first-hop interaction sequence $S_u^\tau$ of node $u$ before timestamp $\tau$ from dynamic graph, where $S_u^\tau = \{(u, k_1, t_1), \ldots, (u, k_{|u|}, t_{|u|})\}$ with $|u|$ denoting the sequence length.

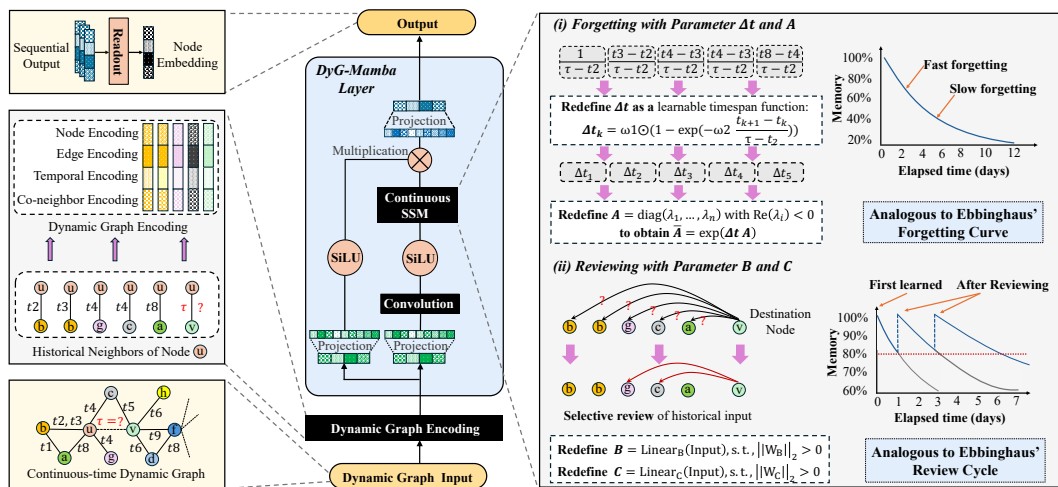

Figure 1: Overview of our proposed DyG-Mamba with four redefined core parameters $\boldsymbol{\Delta}$, $\boldsymbol{A}$, $\boldsymbol{B}$ and $\boldsymbol{C}$. Pseudocodes are in Appendix C.

**Node and Edge Encoding.** We directly adopt the node and edge features provided by datasets as node encoding $\boldsymbol{X}_{u,V}^\tau \in \mathbb{R}^{|u| \times d_N}$ and edge encoding $\boldsymbol{X}_{u,E}^\tau \in \mathbb{R}^{|u| \times d_E}$ for $S_u^\tau$, respectively. If the graph is non-attributed, we simply set the node or edge encoding to zero vectors.

**Absolute Temporal Encoding.** We encode the absolute timespans between timestamp $t_j$ and the final prediction timestamp $\tau$ by using an encoding function $\cos(\boldsymbol{\omega}(\tau - t_j))$ to obtain the absolute temporal encoding $\boldsymbol{X}_{u,T}^\tau \in \mathbb{R}^{|u| \times d_T}$, where $\boldsymbol{\omega} = \{\alpha^{-(i-1)/\beta}\}_{i=1}^{d_T}$ with $\alpha$ and $\beta$ as trainable parameters. Following [9], we keep $\boldsymbol{\omega}$ constant during training to facilitate easier model optimization.

**Co-occurrence Frequency Encoding.** Two nodes that frequently interact with the same neighbors tend to have similar embeddings. Thus, we capture this feature by adopting co-occurrence frequency encoding. Formally, let the neighbors of $u$ and $v$ be $S_u = \{a, b\}$ and $S_v = \{b, b, c, a\}$, the co-occurrence features of $u$ could be denoted by $\boldsymbol{C}_u^\tau = [[1,1], [1,2]]$, where $[1,1]$ denotes the occurrence frequency of $a$ in $S_a$ and $S_b$, respectively. Then, we define a function $f(\cdot) : \mathbb{R}^1 \to \mathbb{R}^{d_C}$ to encode the co-occurrence features by:

$$\boldsymbol{X}_{u,C}^\tau = (f(\boldsymbol{C}_u^\tau[:,0]) + f(\boldsymbol{C}_u^\tau[:,1]))\boldsymbol{W}_C + \boldsymbol{b}_C, \tag{5}$$

where $\boldsymbol{X}_{u,C}^\tau \in \mathbb{R}^{|u| \times d_C}$ with $d_C$ denotes dimensionality, $\boldsymbol{W}_C$ and $\boldsymbol{b}_C$ are trainable parameters. We implement $f(\cdot) : \mathbb{R}^1 \to \mathbb{R}^{d_C}$ by two-layer perception with ReLU activation.

**Encoding Alignment.** We align the above-mentioned encoding to the same dimension $d$:

$$\boldsymbol{Z}_{u,*}^\tau = \boldsymbol{X}_{u,*}^\tau \boldsymbol{W}_* + \boldsymbol{b}_*, \quad \text{where } * \in \{N, E, T, C\}, \tag{6}$$

where $\boldsymbol{W}_* \in \mathbb{R}^{d_* \times d}$ and $b_* \in \mathbb{R}^d$ are trainable parameters. Finally, we concatenate aligned encoding for $S_u^\tau$ as $\boldsymbol{Z}_u^\tau = \boldsymbol{Z}_{u,N}^\tau \| \boldsymbol{Z}_{u,E}^\tau \| \boldsymbol{Z}_{u,T}^\tau \| \boldsymbol{Z}_{u,C}^\tau$ and $\boldsymbol{Z}_u^\tau \in \mathbb{R}^{|u| \times 4d}$.

## 4.2 DyG-Mamba: Dynamic Graph Mamba

### 4.2.1 Rethinking SSM on Dynamic Graph

To learn node embedding of $u$, SSM first encodes $\boldsymbol{Z}_u^\tau$ using a linear layer followed by a 1D convolution layer and SiLU activation function, which could be formulated by

$$\boldsymbol{M}_u^\tau = \text{SiLU}(\text{Conv1D}(\text{Linear}(\boldsymbol{Z}_u^\tau))) \in \mathbb{R}^{|u| \times 8d}. \tag{7}$$

Then, SSM initializes four core trainable parameters: $\boldsymbol{A} \in \mathbb{R}^{|u| \times 8d \times 8d}$ governs state transition, $\boldsymbol{B} \in \mathbb{R}^{|u| \times 8d}$ and $\boldsymbol{C} \in \mathbb{R}^{|u| \times 8d}$ governs input/output projections, and $\boldsymbol{\Delta t}$ controls system's update step size [19]. And the $k$-th output of the SSM is given by:

$$\boldsymbol{h}_k = \overline{\boldsymbol{A}}_k \boldsymbol{h}_{k-1} + \overline{\boldsymbol{B}}_k \boldsymbol{m}_k, \tag{8}$$

$$\widehat{\boldsymbol{m}}_k^\tau = \overline{\boldsymbol{C}}_k \boldsymbol{h}_k, \tag{9}$$

where $\boldsymbol{m}_k$ represents the $k$-th input, $\overline{\boldsymbol{A}}_k = \exp(\Delta t_k \boldsymbol{A}_k)$, $\overline{\boldsymbol{B}}_k = (\Delta t_k \boldsymbol{A}_k)^{-1}(\overline{\boldsymbol{A}}_k - \boldsymbol{I})(\Delta t_k \boldsymbol{B}_k)$, $\overline{\boldsymbol{C}}_k = \boldsymbol{C}_k$, and $\boldsymbol{h}_k$ denotes node's $k$-th hidden state representation.

Finally, SSM adopts skip connection to avoid gradient vanishing and generates the sequential output:

$$\widehat{\boldsymbol{Z}}_{u,\text{out}}^{\tau} = (\widehat{\boldsymbol{M}}_u^{\tau} \odot \text{SiLU}(\text{Linear}(\boldsymbol{Z}_u^{\tau})))\boldsymbol{W}_{\text{out}} + \boldsymbol{b}_{\text{out}}, \tag{10}$$

where $\odot$ denotes element-wise product, $\boldsymbol{W}_{\text{out}} \in \mathbb{R}^{8d \times 4d}$ and $\boldsymbol{b}_{\text{out}} \in \mathbb{R}^{4d}$ are trainable parameters.

As shown in Eq.(8,9), SSM contains three core parameters, $\overline{\boldsymbol{A}}_k$, $\overline{\boldsymbol{B}}_k$ and $\overline{\boldsymbol{C}}_k$, which determine the effectiveness for long-term sequence modeling. Specifically, (i) $\overline{\boldsymbol{A}}_k$ controls the forgetting of historical information, determined by $\Delta t_k$ and $\boldsymbol{A}_k$. Existing SSMs, such as Hippo [39] and Mamba, typically initialize $\boldsymbol{A}_k$ randomly and either fix the step size $\Delta \boldsymbol{t}$ as a constant or adopt a data-dependent strategy to set $\Delta \boldsymbol{t} = \text{SiLU}(\text{Linear}(\boldsymbol{Z}_u^{\tau}))$. However, these SSMs do not account for the crucial role of irregular timespans in real-world sequential input, leading to suboptimal performance. Moreover, directly tying the input $\boldsymbol{Z}_u^{\tau}$ to $\Delta \boldsymbol{t}$ further weakens SSM's effectiveness and inductiveness, as it will encounter a large variety of unseen input during testing. (ii) Existing SSMs struggle to effectively filter out noisy historical information. Although Mamba initializes $\boldsymbol{B}$ and $\boldsymbol{C}$ as data-dependent parameters, allowing $\overline{\boldsymbol{B}}_k$ and $\overline{\boldsymbol{C}}_k$ to selectively copy important past information, input noise can still affect their initialization, thereby weakening their robustness [17]. To address these issues, we propose DyG-Mamba, a timespan-informed continuous SSM designed for dynamic graph modeling.

### 4.2.2 Timespan-Informed Continuous SSM

To better utilize irregular timespans and enhance the effectiveness and inductiveness of SSMs, we first redefine $\Delta \boldsymbol{t}$ and $\boldsymbol{A}$. Ebbinghaus's forgetting curve describes how memory retention decreases exponentially over time and can be formulated as $R = \exp(-t/S)$ [40, 41], where $R$ denotes memory retention, $t$ is the time interval, and $S$ is a decay constant. Inspired by this formulation, we reinterpret $R$ as a timespan-dependent decay coefficient, enabling the model to apply temporal decay to historical states proportionally to the elapsed timespan. Accordingly, we design DyG-Mamba with a similar exponential forgetting mechanism, where the core parameter $\overline{\boldsymbol{A}}_k$, which governs the degree of forgetting, decays exponentially as the $k$-th timespan $(t_{k+1} - t_k)$ increases. Since $\overline{\boldsymbol{A}}_k$ is jointly determined by both $\Delta \boldsymbol{t}$ and $\boldsymbol{A}$, this mechanism is realized by redefining these two variables.

**Redefining Parameter $\Delta \boldsymbol{t}$.** To establish the forgetting curve relationship between timespans and $\overline{\boldsymbol{A}}_k$, we first define the connection between timespans and the step size $\Delta \boldsymbol{t}$. Since $\Delta \boldsymbol{t}$ could directly influence $\overline{\boldsymbol{A}}_k$. Specifically, we define a monotonically increasing, learnable timespan function to redefine the step size parameter $\Delta \boldsymbol{t}$ as

$$\Delta t_k = \boldsymbol{w}_1 \odot \left(\boldsymbol{1} - \exp\left(-\boldsymbol{w}_2 \odot \frac{t_{k+1} - t_k}{\tau - t_1}\right)\right), \tag{11}$$

where $\Delta t_k$ denotes the $k$-th step size, $(t_{k+1} - t_k)$ denotes the $k$-th timespan, $\boldsymbol{w}_1 \in \mathbb{R}^{8d}$ and $\boldsymbol{w}_2 \in \mathbb{R}^{8d}$ are trainable vectors designed to capture fine-grained timespan features, with each element constrained to be positive. $\boldsymbol{1}$ is an all-ones vector, $\odot$ denotes element-wise multiplication, and $\tau$ and $t_1$ are the last and first appearing timestamps, respectively.

This redefinition of $\Delta \boldsymbol{t}$ establishes a direct relationship between timespan length and step size scaling. Its monotonically increasing property ensures that longer timespans induce stronger decay in $\overline{\boldsymbol{A}}_k = \exp(\Delta t_k \boldsymbol{A}_k)$. Consequently, careful initialization of $\boldsymbol{A}$ is required to maintain a balance between effective forgetting and numerical stability.

**Redefining Parameter $\boldsymbol{A}$.** We consider two factors when redefining the initialization of $\boldsymbol{A}$. (i) $\boldsymbol{A}$ should maintain the forgetting curve relationship, *i.e.*, $\exp(\Delta t_k \boldsymbol{A}_k)$ should diminish exponentially as $\Delta t_k$ increases. (ii) $\boldsymbol{A}$ determines the stability of recurrent updating in long-term sequence modeling [42], *i.e.*, it should prevent gradient vanishing or explosion over time. To satisfy these two requirements, we initialize $\boldsymbol{A}$ as a diagonal matrix whose eigenvalues all have negative real parts. Theorem 4.1 confirms that this initialization strategy satisfies both conditions.

**Theorem 4.1.** *Let* $\boldsymbol{A}_k = \text{diag}(\lambda_1, \ldots, \lambda_n)$, *where the real parts of the eigenvalues satisfy* $\text{Re}(\lambda_i) < 0$. *For any timespan* $\Delta t_k$, *we have* $\overline{\boldsymbol{A}}_k = \text{diag}(e^{\lambda_1 \Delta t_{k,1}}, \ldots, e^{\lambda_n \Delta t_{k,n}})$ *and* $\overline{\boldsymbol{B}}_k = \text{diag}(\lambda_1^{-1}(e^{\lambda_1 \Delta t_{k,1}} - 1)\boldsymbol{B}_{k,1}, \ldots, \lambda_n^{-1}(e^{\lambda_n \Delta t_{k,n}} - 1)\boldsymbol{B}_{k,n})$, *where* $\Delta t_{k,i}$ *and* $\boldsymbol{B}_{k,i}$ *are the* $i$-*th elements of* $\Delta t_k$ *and* $\boldsymbol{B}_k$. *The* $i$-*th coordinate of* $\boldsymbol{h}_k$ *is denoted as* $\boldsymbol{h}_{k,i} = e^{\lambda_i \Delta t_{k,i}} \boldsymbol{h}_{k-1,i} + \lambda_i^{-1}(e^{\lambda_i \Delta t_{k,i}} - 1)\boldsymbol{B}_{k,i} \boldsymbol{m}_{k,i}$.

(i) Theorem 4.1 guarantees the forgetting curve relationship. When $\Delta t_{k,i}$ is sufficiently small, then $e^{\lambda_i \Delta t_{k,i}} \approx 1$, *i.e.*, $\boldsymbol{h}_{k,i} \approx \boldsymbol{h}_{k-1,i}$. This indicates that for small timespans, the model retains historical states and disregards the current input. On the other hand, as the timespan increases, $e^{\lambda_i \Delta t_{k,i}}$ gradually approaches 0 at a decreasing rate over time, causing the system to forget previous states and place greater emphasis on current input $\boldsymbol{m}_{k,i}$. (ii) Since $\mathrm{Re}(\lambda_i) < 0$, the term $|e^{\lambda_i \Delta t_{k,i}}|$ is bounded by 1 and decreases as $\Delta t_{k,i}$ increases. This prevents the hidden state from diverging over extended sequences and mitigates gradient explosion, ensuring the stability of recurrent updates.

Traditional SSMs struggle to effectively filter out noise from the input sequence. To solve this issue, we redefine $\boldsymbol{B}$ and $\boldsymbol{C}$ as input-dependent parameters and introduce spectral norm constraints to enhance robustness. Specifically, Ebbinghaus' review cycle indicates that periodic review of previously learned information helps counteract memory decay [20]. Inspired by this, we design DyG-Mamba to continuously review important node while forgetting irrelevant or noisy inputs.

**Redefining Parameter $\boldsymbol{B}$ and $\boldsymbol{C}$.** To align the SSM with the review cycle, we first define $\boldsymbol{B}$ and $\boldsymbol{C}$ as input-dependent parameters, *e.g.*, $\boldsymbol{B} = \mathrm{Linear}_{\mathrm{B}}(\boldsymbol{M}_u^\tau)$. Then we can filter out noise by Theorem 4.2.

**Theorem 4.2.** *Let $\boldsymbol{B}$ and $\boldsymbol{C}$ be input-dependent parameters and $\boldsymbol{m}_k$ denote the $k$-th input in $\boldsymbol{M}_u^\tau$. Update process for SSMs, as shown in Eq.(8), could be further decomposed as*

$$\widehat{\boldsymbol{m}}_k^\tau = \overline{\boldsymbol{C}}_k \prod_{i=0}^{k-2} \overline{\boldsymbol{A}}_{k-i} \overline{\boldsymbol{B}}_1 \boldsymbol{m}_1 + \cdots + \overline{\boldsymbol{C}}_k \overline{\boldsymbol{B}}_k \boldsymbol{m}_k, \tag{12}$$

$$= e^{(\sum_{i=0}^{k-2} \Delta t_{k-i} \boldsymbol{A}_{k-i})} \overline{\boldsymbol{C}}_k \overline{\boldsymbol{B}}_1 \boldsymbol{m}_1 + \cdots + \overline{\boldsymbol{C}}_k \overline{\boldsymbol{B}}_k \boldsymbol{m}_k.$$

According to Theorem 4.2, $\overline{\boldsymbol{C}}_k$ can be interpreted as the query corresponding to the $k$-th input, while $\overline{\boldsymbol{B}}_k$ and $\boldsymbol{m}_k$ serve as the key and value, respectively. Thus, the product $\overline{\boldsymbol{C}}_k \overline{\boldsymbol{B}}_j$ represents the importance of the $j$-th historical input $\boldsymbol{m}_j$ to the $k$-th input. While Theorem 4.2 enables automatic filtering of irrelevant and noisy historical inputs, the construction of $\boldsymbol{B}$ and $\boldsymbol{C}$ using only linear layers makes them susceptible to input noise. To address this issue, we introduce spectral norm constraints on the initialization of $\boldsymbol{B}$ and $\boldsymbol{C}$ to achieve dual objectives, formulated by

$$\boldsymbol{B} = \boldsymbol{W}_{\mathrm{B}} \boldsymbol{M}_u^\tau + \boldsymbol{b}_{\mathrm{B}}, \quad \boldsymbol{C} = \boldsymbol{W}_{\mathrm{C}} \boldsymbol{M}_u^\tau + \boldsymbol{b}_{\mathrm{C}}, \tag{13}$$
$$s.t. \quad \|\boldsymbol{W}_{\mathrm{B}}\|_2 \leq 1, \quad \|\boldsymbol{W}_{\mathrm{C}}\|_2 \leq 1,$$

where $\boldsymbol{W}_{\mathrm{B}}$ and $\boldsymbol{W}_{\mathrm{C}}$ are weight matrices, and $\|\cdot\|_2$ is the spectral norm, *i.e.*, the largest singular value of $\boldsymbol{W}_{\mathrm{B/C}}$. The spectral norm constraints guarantee Lipschitz continuity, ensuring that $\boldsymbol{B}$ and $\boldsymbol{C}$ remain stable under input perturbations. Overall, DyG-Mamba is robust to input noise, preventing irrelevant samples from being erroneously reinforced. This robustness is guaranteed by Theorem 4.3.

**Theorem 4.3.** *Given $\|\boldsymbol{W}_B\|_2 \leq 1$, $\|\boldsymbol{W}_C\|_2 \leq 1$, $\gamma = \max_i \mathrm{Re}(\lambda_i(\boldsymbol{A})) < 0$, and $T$ as the total sequence duration, the output perturbation satisfies:*

$$\|\Delta \widehat{\boldsymbol{m}}_k^\tau\| \leq \frac{1}{|\gamma|} \left(1 - e^{\gamma T}\right) \|\Delta \boldsymbol{M}_u^\tau\|. \tag{14}$$

### 4.3 DyG-Mamba for Downstream Tasks

For dynamic link prediction, we first process the first-hop interaction sequences of source node $u$ and destination node $v$ through two independent DyG-Mamba models. Based on Eq.(7-10), two models generate sequential output embeddings $\widehat{\boldsymbol{Z}}_{u,\mathrm{out}}^\tau$ and $\widehat{\boldsymbol{Z}}_{v,\mathrm{out}}^\tau$, respectively. Then, we adopt readout function, *i.e.*, MEAN pooling, to obtain their node embedding, defined as $\widehat{\boldsymbol{z}}_*^\tau = \mathrm{MEAN}(\widehat{\boldsymbol{Z}}_{*,\mathrm{out}}^\tau)$ with $* \in \{u, v\}$. Finally, we concatenate two node embedding and adopt an MLP for link prediction:

$$\hat{y} = \mathrm{Signoid}(\mathrm{Linear}(\mathrm{ReLU}(\mathrm{Linear}(\widehat{\boldsymbol{z}}_u^\tau \| \widehat{\boldsymbol{z}}_v^\tau)))). \tag{15}$$

We adopt binary cross-entropy loss for optimization

$$\mathcal{L}_{\mathrm{LP}} = -\frac{1}{|\mathcal{B}|} \sum_{i=1}^{|\mathcal{B}|} \left[y_i \log \hat{y}_i + (1 - y_i) \log(1 - \hat{y}_i)\right], \tag{16}$$

where $|\mathcal{B}|$ denotes the batch size containing both positive and negative samples, and $y_i$ and $\hat{y}_i$ represent the $i$-th ground-truth and predicted label, respectively.

For dynamic node classification, we use one DyG-Mamba model and discard co-occurrence encoding, while keeping other components identical to the link prediction setup.

**Computational Complexity.** Given batch size $b$, feature dimension $d$, and sequence length $L$. DyG-Mamba achieves linear memory and time complexity of $\mathcal{O}(bdL)$, while DyGFormer has quadratic complexity of $O(bdL^2)$. This highlights the efficiency of DyG-Mamba. Details in Appendix B.

Table 2: **Transductive**: AP for dynamic link prediction with random (*rnd*), historical (*hist*), and inductive (*ind*) negative edge sampling. **bold** and underlined emphasize best and 2nd-best results.

| | Datasets | JODIE | DyRep | TGN | CAWN | TGAT | EdgeBank | GraphMixer | TCL | DyGFormer | DyG-Mamba |
|---|---|---|---|---|---|---|---|---|---|---|---|
| *rnd* | Wikipedia | 96.50±0.14 | 94.86±0.06 | 98.45±0.06 | 98.76±0.03 | 96.94±0.06 | 90.37±0.00 | 97.25±0.03 | 96.47±0.16 | 99.03±0.02 | **99.06±0.01** |
| | Reddit | 98.31±0.14 | 98.22±0.04 | 98.63±0.06 | 99.11±0.01 | 98.52±0.02 | 94.86±0.00 | 97.31±0.01 | 97.53±0.02 | 99.22±0.01 | **99.25±0.00** |
| | MOOC | 80.23±2.44 | 81.97±0.49 | 89.15±1.60 | 80.15±0.25 | 85.84±0.15 | 57.97±0.00 | 82.78±0.15 | 82.38±0.24 | 87.52±0.49 | **90.17±0.19** |
| | LastFM | 70.85±2.13 | 71.92±2.21 | 77.07±3.97 | 86.99±0.06 | 73.42±0.21 | 79.29±0.00 | 75.61±0.24 | 67.27±2.16 | 93.00±0.12 | **94.22±0.04** |
| | Enron | 84.77±0.30 | 82.38±3.36 | 86.53±1.11 | 89.56±0.09 | 71.12±0.97 | 83.53±0.00 | 82.25±0.16 | 79.70±0.71 | 92.47±0.12 | **93.22±0.03** |
| | Social Evo. | 89.89±0.55 | 88.87±0.30 | 93.57±0.17 | 84.96±0.09 | 93.16±0.17 | 74.95±0.00 | 93.37±0.07 | 93.13±0.16 | 94.73±0.01 | **94.75±0.01** |
| | UCI | 89.43±1.09 | 65.14±2.30 | 92.34±1.04 | 95.18±0.06 | 79.63±0.70 | 76.20±0.00 | 93.25±0.57 | 89.57±1.63 | 95.79±0.17 | **96.79±0.08** |
| | Can. Parl. | 69.26±0.31 | 66.54±2.76 | 70.88±2.34 | 69.82±2.34 | 70.73±0.72 | 64.55±0.00 | 77.04±0.46 | 68.67±2.67 | 97.36±0.45 | **98.37±0.07** |
| | US Legis. | 75.05±1.52 | 75.34±0.39 | **75.99±0.58** | 70.58±0.48 | 68.52±3.16 | 58.39±0.00 | 70.74±1.02 | 69.59±0.48 | 71.11±0.59 | 74.11±2.32 |
| | UN Trade | 64.94±0.31 | 63.21±0.93 | 65.03±1.37 | 65.39±0.12 | 61.47±0.18 | 60.41±0.00 | 62.61±0.27 | 62.21±0.03 | 66.46±1.29 | **68.55±0.16** |
| | UN Vote | 63.91±0.81 | 62.81±0.80 | 65.72±2.17 | 52.84±0.10 | 52.21±0.98 | 58.49±0.00 | 52.11±0.16 | 51.90±0.30 | 55.55±0.42 | **65.69±1.10** |
| | Contact | 95.31±1.33 | 95.98±0.15 | 96.89±0.56 | 90.26±0.28 | 96.28±0.09 | 92.58±0.00 | 91.92±0.03 | 92.44±0.12 | 98.29±0.01 | **98.37±0.01** |
| | Avg. Rank | 6.08 | 6.00 | 4.42 | 8.42 | 6.33 | 6.92 | 4.92 | 6.58 | 2.92 | **2.42** |
| *hist* | Wikipedia | 83.01±0.66 | 79.93±0.56 | 86.86±0.33 | 71.21±1.67 | 87.38±0.22 | 73.35±0.00 | **90.90±0.10** | 89.05±0.39 | 82.23±2.54 | 82.12±1.22 |
| | Reddit | 80.03±0.36 | 79.83±0.31 | 81.22±0.61 | 80.82±0.45 | 79.55±0.20 | 73.59±0.00 | 78.44±0.18 | 77.14±0.16 | **81.57±0.67** | 81.16±0.11 |
| | MOOC | 78.94±1.25 | 75.60±1.12 | 87.06±1.93 | 74.05±0.95 | 82.19±0.62 | 60.71±0.00 | 77.77±0.92 | 77.06±0.41 | 85.85±0.66 | **87.33±1.46** |
| | LastFM | 74.35±3.81 | 74.92±2.46 | 76.87±4.64 | 69.86±0.43 | 71.59±0.24 | 73.03±0.00 | 72.47±0.49 | 59.30±2.31 | 81.57±0.48 | **84.09±0.44** |
| | Enron | 69.85±2.70 | 71.19±2.76 | 73.91±1.76 | 64.73±0.36 | 64.07±1.05 | 76.53±0.00 | **77.98±0.92** | 70.66±0.39 | 75.63±0.73 | 77.41±1.13 |
| | Social Evo. | 87.44±6.78 | 93.29±0.43 | 94.45±0.56 | 85.53±0.38 | 95.01±0.44 | 80.57±0.00 | 94.93±0.31 | 94.74±0.31 | **97.38±0.14** | 96.59±0.28 |
| | UCI | 75.24±5.80 | 55.10±3.14 | 80.43±2.12 | 65.30±0.43 | 68.27±1.37 | 65.50±0.00 | **84.11±1.35** | 80.25±2.74 | 82.17±0.82 | 82.95±2.24 |
| | Can. Parl. | 51.79±0.63 | 63.31±1.23 | 68.42±3.07 | 66.53±2.77 | 67.13±0.84 | 63.84±0.00 | 74.34±0.87 | 65.93±3.00 | 97.00±0.31 | **97.22±0.29** |
| | US Legis. | 51.71±5.76 | 86.88±2.25 | 74.00±7.57 | 68.82±8.23 | 62.14±6.60 | 63.22±0.00 | 81.65±1.02 | 80.53±3.95 | 85.30±3.88 | **88.83±0.34** |
| | UN Trade | 61.39±1.83 | 59.19±1.07 | 58.44±5.51 | 55.71±0.38 | 55.74±0.91 | **81.32±0.00** | 57.05±1.22 | 55.90±1.17 | 64.41±1.40 | 65.19±0.19 |
| | UN Vote | 70.02±0.81 | 69.30±1.12 | 69.37±3.93 | 51.26±0.04 | 52.96±2.14 | **84.89±0.00** | 51.20±1.60 | 52.30±2.35 | 60.84±1.58 | 59.51±3.08 |
| | Contact | 95.31±2.13 | 96.39±0.20 | 93.05±2.35 | 84.16±0.49 | 96.05±0.52 | 88.81±0.00 | 93.36±0.41 | 93.86±0.21 | 97.57±0.06 | **97.80±0.14** |
| | Avg. Rank | 6.08 | 6.00 | 4.42 | 8.42 | 6.33 | 6.92 | 4.92 | 6.58 | 3.00 | **2.33** |
| *ind* | Wikipedia | 75.65±0.79 | 70.21±1.58 | 85.62±0.44 | 74.06±2.62 | 87.00±0.16 | 80.63±0.00 | **88.59±0.17** | 86.76±0.72 | 78.29±5.38 | 84.64±0.77 |
| | Reddit | 86.98±0.16 | 86.30±0.26 | 88.10±0.24 | 91.67±0.24 | 89.59±0.24 | 85.48±0.00 | 85.26±0.11 | 87.45±0.29 | 91.11±0.40 | **91.89±0.42** |
| | MOOC | 65.23±2.19 | 61.66±0.95 | 77.50±2.91 | 73.51±0.94 | 75.95±0.64 | 49.43±0.00 | 74.27±0.92 | 74.65±0.54 | **81.24±0.69** | 81.15±1.25 |
| | LastFM | 62.67±4.49 | 64.41±2.70 | 65.95±5.98 | 67.48±0.77 | 71.13±0.17 | **75.49±0.00** | 68.12±0.33 | 58.21±0.89 | 73.97±0.50 | 74.76±0.40 |
| | Enron | 68.96±0.98 | 67.79±1.53 | 70.89±2.72 | 75.15±0.58 | 63.94±1.36 | 73.89±0.00 | 75.01±0.79 | 71.29±0.32 | 77.41±0.89 | **79.90±0.90** |
| | Social Evo. | 89.82±4.11 | 93.28±0.48 | 95.13±0.56 | 88.32±0.27 | 94.84±0.44 | 83.69±0.00 | 94.72±0.33 | 94.90±0.36 | **97.68±0.10** | 96.91±0.24 |
| | UCI | 65.99±1.40 | 54.79±1.76 | 70.94±0.71 | 64.61±0.48 | 68.67±0.84 | 57.43±0.00 | **80.10±0.51** | 76.01±1.11 | 72.25±1.71 | 73.71±3.88 |
| | Can. Parl. | 48.42±0.66 | 58.61±0.86 | 65.34±2.87 | 67.75±1.00 | 68.82±1.21 | 64.74±0.00 | 69.48±0.63 | 65.85±1.75 | 95.44±0.57 | **96.58±0.79** |
| | US Legis. | 50.27±5.13 | 83.44±1.16 | 67.57±6.47 | 65.81±8.52 | 61.91±5.82 | 64.74±0.00 | 79.63±0.84 | 78.15±3.34 | 81.25±3.62 | **85.03±0.69** |
| | UN Trade | 60.42±1.48 | 60.19±1.24 | 61.04±6.01 | 62.54±0.67 | 60.61±1.24 | **72.97±0.00** | 60.15±1.29 | 61.06±1.74 | 55.79±1.02 | 61.88±1.46 |
| | UN Vote | 67.79±1.46 | **67.53±1.98** | 67.63±2.67 | 52.19±0.34 | 52.89±1.61 | 66.30±0.00 | 51.60±0.73 | 50.62±0.82 | 51.91±0.84 | 57.63±1.15 |
| | Contact | 93.43±1.78 | 94.18±0.10 | 90.18±3.28 | 89.31±0.27 | 94.35±0.48 | 85.20±0.00 | 90.87±0.35 | 91.35±0.21 | **94.75±0.28** | 94.57±0.22 |
| | Avg. Rank | 7.33 | 7.25 | 5.17 | 6.17 | 5.25 | 6.75 | 5.42 | 5.58 | 3.75 | **2.33** |

## 5 Experiments

### 5.1 Experimental Setup

**Datasets and Baselines**. We evaluate performance on 12 datasets, each split into 70%/15%/15% for training, validation and testing. Details in Appendix D.1. We select nine SOTA baselines for comparison, *e.g.*, four RNN-based methods: JODIE [2], DyRep [29], TGN [5] and CAWN [11], a GNN-based method: TGAT [11], a memory-based method: EdgeBank [30], a MLP-based method: GraphMixer [9], and two Transformer-based methods: TCL [23] and DyGFormer [6].

**Implementation Details**. For a fair comparison, we use DyGLib [6] to reproduce all baselines via the same training and inference pipeline. We set the same input length for DyGFormer and DyG-Mamba to fairly compare the long-term dynamic graph modeling ability. We train each model for 100 epochs and select the best-performing checkpoint for testing. We repeat each experiment 10 times with different random seeds and report the mean and standard derivation. Details in Appendix D.2.

**Evaluation Details**. We evaluate baselines in transductive and inductive settings, where the former predicts future links among nodes seen during training, and the latter focus on unseen nodes [6]. For negative sampling, we follow [30] and adopt random (*rnd*), historical (*hist*), and inductive (*ind*) strategies (see Appendix D.3). Metrics are Average Precision (AP) and AUC-ROC.

Table 3: **Inductive**: AP for *dynamic link prediction* with random negative edge sampling strategies. The notations are the same as Table 2.

| Datasets | JODIE | DyRep | TGN | CAWN | TGAT | TCL | GraphMixer | DyGFormer | DyG-Mamba |
|---|---|---|---|---|---|---|---|---|---|
| Wikipedia | 94.82±0.20 | 92.43±0.37 | 97.83±0.04 | 98.24±0.03 | 96.22±0.07 | 96.65±0.02 | 96.22±0.17 | 98.59±0.03 | **98.66±0.02** |
| Reddit | 96.50±0.13 | 96.09±0.11 | 97.50±0.07 | 98.62±0.01 | 97.09±0.04 | 95.26±0.02 | 94.09±0.07 | 98.84±0.02 | **98.91±0.01** |
| MOOC | 79.63±1.92 | 81.07±0.44 | 89.04±1.17 | 81.42±0.24 | 85.50±0.19 | 81.41±0.21 | 80.60±0.22 | 86.96±0.43 | **89.98±0.04** |
| LastFM | 81.61±3.82 | 83.02±1.48 | 81.45±4.29 | 89.42±0.07 | 78.63±0.31 | 82.11±0.42 | 73.53±1.66 | 94.23±0.09 | **95.16±0.05** |
| Enron | 80.72±1.39 | 74.55±3.95 | 77.94±1.02 | 86.35±0.51 | 67.05±1.51 | 75.88±0.48 | 76.14±0.79 | 89.76±0.34 | **90.97±0.01** |
| Social Evo. | 91.96±0.48 | 90.04±0.47 | 90.77±0.86 | 79.94±0.18 | 91.41±0.16 | 91.86±0.06 | 91.55±0.09 | 93.14±0.04 | **93.17±0.05** |
| UCI | 79.86±1.48 | 57.48±1.87 | 88.12±2.05 | 92.73±0.06 | 79.54±0.48 | 91.19±0.42 | 87.36±2.03 | **94.54±0.12** | 93.38±0.21 |
| Can. Parl. | 53.92±0.94 | 54.02±0.76 | 54.10±0.93 | 55.80±0.69 | 55.18±0.79 | 55.91±0.82 | 54.30±0.66 | 87.74±0.71 | **96.64±0.04** |
| US Legis. | 54.93±2.29 | 57.28±0.71 | **58.63±0.37** | 53.17±1.20 | 51.00±3.11 | 50.71±0.76 | 52.59±0.97 | 54.28±2.87 | 55.25±4.54 |
| UN Trade | 59.65±0.77 | 57.02±0.69 | 58.31±3.15 | 65.24±0.21 | 61.03±0.18 | 62.17±0.31 | 62.21±0.12 | 64.55±0.62 | **67.04±0.20** |
| UN Vote | 56.64±0.96 | 54.62±2.22 | **58.85±2.51** | 49.94±0.45 | 52.24±1.46 | 50.68±0.44 | 51.60±0.97 | 55.93±0.39 | 58.08±0.55 |
| Contact | 94.34±1.45 | 92.18±0.41 | 93.82±0.99 | 89.55±0.30 | 95.87±0.11 | 90.59±0.05 | 91.11±0.12 | 98.03±0.02 | **98.10±0.02** |
| Avg. Rank | 5.83 | 6.83 | 4.67 | 4.92 | 6.21 | 6.00 | 6.71 | 2.50 | **1.33** |

## 5.2 Effectiveness Evaluation

Table 2 and Table 3 show models' performance in dynamic link prediction under transductive and inductive settings. AUC-ROC score is reported in the Appendix D.4. From these tables, we observe that DyG-Mamba achieves the best performance on most datasets and achieves the best average rank in both AP and AUC-ROC across three negative edge sampling strategies, demonstrating its higher effectiveness and better generalization compared to SOTA baselines. The primary reasons for DyG-Mamba's superior performance can be summarized in three key aspects. **(i)**. DyG-Mamba employs an SSM architecture that effectively handles long-term sequences. In contrast, DyGFormer requires patching under the same input length, which compresses the sequence data and leads to information loss. **(ii)**. DyG-Mamba leverages irregular temporal information to control the compression of historical states, thereby making more efficient use of time information and enhancing model's generalization capability. **(iii)**. DyG-Mamba selectively filters out past noise or irrelevant information, resulting in more robust node embedding and improved prediction accuracy.

To further verify the scalability and efficiency of DyG-Mamba on million-edge temporal graphs, we conducted additional experiments on `tgbl-coin-v2` [43], which contains 638K nodes and 22.8M temporal edges. Following the standardized training pipeline and hyperparameter setup in Yu et al. [44], we ensured a fair comparison with the existing baselines. As summarized in Table 4, DyG-Mamba not only achieves superior predictive performance but also demonstrates remarkable training efficiency, further validating its scalability on real-world large-scale dynamic graphs.

Table 4: Scalability on million-edge temporal graphs.

| Method | Performance | Running Time | GPU Usage |
|---|---|---|---|
| DyRep | 45.20±4.60 | 49:38:39 | 48116M |
| TGN | 58.60±3.70 | 38:26:48 | 48116M |
| GraphMixer | 75.31±0.21 | **11:59:20** | **12204M** |
| DyGFormer | 75.17±0.38 | 45:19:11 | 41348M |
| DyG-Mamba | **75.17±0.38** | 12:32:04 | 18094M |

**Scalability of Effectiveness**. As shown in Figure 2, to highlight DyG-Mamba's ability to capture long-term temporal dependencies, we compare it to three best-performing baselines. Obviously, DyG-Mamba's performance improves substantially with longer sequences and outperforms baselines even at shorter sequence lengths, showing its effectiveness in modeling long-term dependencies on dynamic graphs.

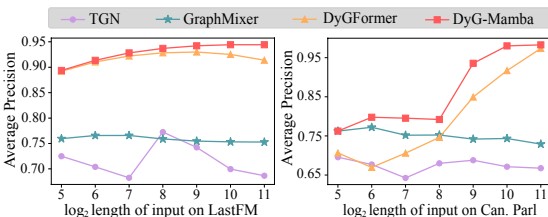
Figure 2: AP score w.r.t. varying sequence lengths.

**Ablation Study**. In Table 5, we conduct an ablation study on three datasets to evaluate the effectiveness of each component in DyG-Mamba. Specifically, we examine five variants: **(i)**. [w/o timespan] replaces timespan with input sample as control signals, *i.e.*, the same setting with vanilla Mamba with $\Delta t = \text{SiLU}(\text{Linear}(\boldsymbol{u}(t)))$. **(ii)**. [w/o Time-encoding] removes the absolute temporal en-

Table 5: Results (AP) of time information ablations.

| Settings | Can. Parl. | Enron | USLegis. |
|---|---|---|---|
| w/o timespan | 96.90±0.18 | 92.14±0.12 | 72.26±0.76 |
| w/o Time-encoding | 97.80±0.43 | 92.83±0.06 | 73.33±1.15 |
| w/o Time | 96.87±0.18 | 92.08±0.12 | 72.19±0.72 |
| w/o Selective | 96.32±0.16 | 91.33±0.10 | 71.62±0.77 |
| w/o Data-dependent | 79.24±0.58 | 82.25±0.16 | 70.57±0.84 |
| DyG-Mamba | **98.37±0.07** | **93.22±0.03** | **74.11±2.23** |

coding $\boldsymbol{X}^{\tau}_{u,T}$. **(iii)**. [w/o Time] removes both timespan and time-encoding. **(iv)**. [w/o Selective] follows parameter settings of S4 [45], *i.e.*, meaning all parameters are independent of input or timespan. **(v)**. [w/o Data-dependent] changes the parameters $\boldsymbol{B}$ and $\boldsymbol{C}$ from being data-dependent to timespan dependent without spectral norm constraints. We observe that removing any component from DyG-Mamba adversely affects its dynamic graph learning capability. Specifically, [w/o timespan] significantly decreases the performance, as it is crucial for capturing irregular temporal patterns. And [w/o Time-encoding] leads to a slight decline in performance since absolute temporal information is also important. Furthermore, [w/o Selective] also degrades performance since the fixed parameters fail to filter out irrelevant noise. Finally, relying entirely on timespan [w/o Data-dependent] also reduces performance, showing the importance of input-dependent setting for parameters $\boldsymbol{B}$ and $\boldsymbol{C}$.

**Effect of Learnable $\Delta t$ Function.** The design of the learnable function for $\Delta t$ in Eq. (11) is inspired by the Ebbinghaus forgetting curve, which models memory retention as $R = \exp(-t/S)$, where $t$ is the elapsed time and $S$ a decay constant. We reinterpret this formulation as a timespan-dependent decay coefficient, ensuring that longer intervals induce stronger decay consistent with human memory dynamics. To validate this choice, we compare several monotonic alternatives, including Linear, Logarithmic, Sigmoid, and

Table 6: Ablation on learnable $\Delta t$ function.

| Variants | Can.Parl. | Enron | USLegis. |
|---|---|---|---|
| Linear | 94.64±0.18 | 91.13±0.06 | 70.28±1.84 |
| Logarithmic | 97.18±0.12 | 92.35±0.05 | 72.46±2.34 |
| Sigmoid | 95.84±0.13 | 92.26±0.04 | 72.15±2.36 |
| Exponential | 94.68±0.11 | 90.84±0.06 | 71.23±1.48 |
| w/o $\tau - t_1$ | 97.32±0.20 | 92.46±0.16 | 72.45±0.84 |
| w/o timespan | 96.90±0.18 | 92.14±0.12 | 72.26±0.76 |
| DyG-Mamba | **98.37±0.07** | **93.22±0.03** | **74.11±2.23** |

Exponential variants, as well as versions without normalization or timespan inputs. As summarized in Table 6, our formulation consistently achieves the best performance across datasets, demonstrating a balanced and smooth decay behavior. In contrast, Linear and Log variants lack boundedness, Sigmoid saturates early, and the Exp variant over-amplifies long timespans, leading to unstable training.

## 5.3 Efficiency Evaluation

Given an input length of 256, Figure 3 and Appendix D.4 show the training time per epoch and the size of trainable parameters on Enron data. Obviously, CAWN requires the longest training time and a substantial number of parameters, since it conducts random walks on dynamic graphs to collect time-aware sequences. In contrast, simpler methods, *e.g.*, GraphMixer and JODIE, have fewer parameters, but exhibit a significant performance gap compared to DyG-Former and DyG-Mamba. Overall, DyG-

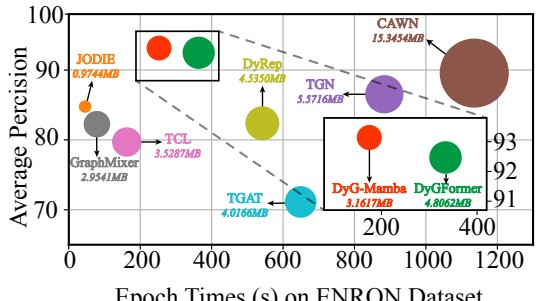

Figure 3: Comparison of efficiency and effectiveness.

Mamba achieves the best performance with a small number of trainable parameters and a moderate training time required per epoch.

**Scalability of Efficiency**. In Figure 4, to highlight DyG-Mamba's ability to effectively capture long-term temporal dependency on dynamic graphs, we provide a more detailed efficiency comparison between Transformer-based DyG-Former and DyG-Mamba. For a fair comparison, we make the same experimental setting for both frameworks. We observe that, with increasing input sequence length, DyG-Mamba demon-

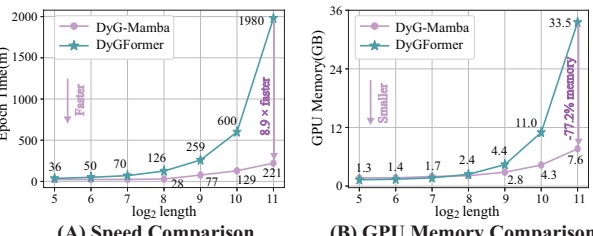

Figure 4: Speed and memory comparison of two layers DyGFormer and DyG-Mamba with varying lengths.

strates a linear growth trend in both runtime and memory consumption, highlighting its efficiency. Specifically, DyG-Mamba is 8.9 times faster than DyGFormer and reduces GPU memory consumption by 77.2% at a sequence length of 2,048. This is because DyG-Mamba only needs a few parameters to compress hidden state and adopts hardware-aware parallel scanning for training.

Table 7: Training convergence comparison on `Can.Parl.` dataset.

| Model | Epoch=20 | Epoch=40 | Epoch=60 | Epoch=80 | Epoch=100 |
|---|---|---|---|---|---|
| Vanilla Mamba | 0.2229 | 0.2026 | 0.1964 | 0.1922 | 0.1914 |
| DyG-Mamba | **0.1691** | **0.1478** | **0.1480** | **0.1482** | **0.1480** |

## 5.4 Training Efficiency

To ensure stable and efficient training, DyG-Mamba incorporates several lightweight yet effective designs. The learnable $\Delta t$ function is implemented as an element-wise exponential decay through a scalar-wise MLP, enabling adaptive modeling of irregular intervals with negligible overhead. A spectral norm constraint regularizes the $B$ and $C$ matrices without adding parameters, preventing instability and ensuring bounded outputs under input noise as supported by Theorem 4.3. Moreover, redefining $B$ and $C$ as input-dependent linear mappings introduces minimal cost while improving the model's adaptability. As shown in Table 7, DyG-Mamba converges smoothly within 100 epochs on the `Can.Parl.` dataset, confirming its fast and stable optimization behavior across datasets.

## 5.5 Robustness Evaluation

We conduct a robustness test by randomly inserting 10% to 60% noisy edges with chronological timestamps during the evaluation. In Figure 5, when the proportion of noisy edges increases, DyG-Mamba exhibits only a minor performance decline, indicating stronger noise robustness compared to baselines. We attribute robustness to the review-based selective memory enhancement, which enables the model to identify most relevant information and filter out noise.

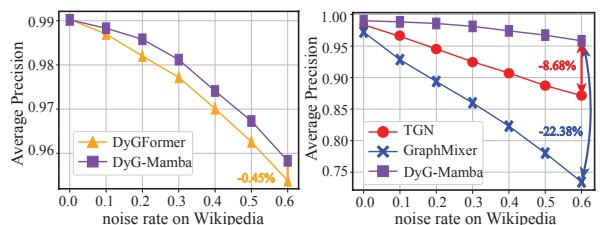

Figure 5: Inserting noisy edges from 10% to 60%.

## 5.6 Case Study

In Figure 6, we randomly extract a middle part sample from one long-term input sequence of the Wikipedia dataset, *e.g.*, $\{1713, 160, 1667, 1667, 1667, 1713, 1667\}$, to visualize DyG-Mamba's efficiency capability for long-term sequence modeling. Figures 6(A)-(B) show the normalized cosine similarity of node embedding between the last and second-last hidden states. We observe that DyGFormer shows high diagonal similarity and nearly uniform similarity across all neighbors, indicat-

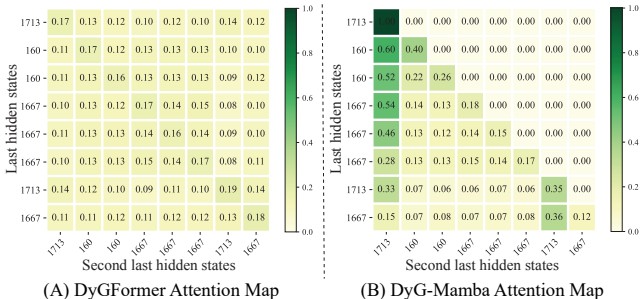

(A) DyGFormer Attention Map     (B) DyG-Mamba Attention Map

Figure 6: Investigate sequential modeling via attention map between source and destination nodes on link prediction.

ing that it struggles to distinguish important historical information. In contrast, DyG-Mamba assigns greater weights to the historical reappearing destination nodes, better enhancing the node embedding and filtering out irrelevant and noisy historical information.

## 6 Conclusion

In this work, we propose a novel SSM framework called DyG-Mamba to effectively and efficiently capture long-term temporal dependencies on dynamic graphs. To achieve this goal, we incorporate irregular time spans as controllable signals, thus establishing a strong correlation between dynamic evolution patterns and time information. We also implement a review-based selective memory enhancement to further improve the model's robustness. Experimental evaluations on various downstream tasks show DyG-Mamba's higher performance and better robustness. In the future, we plan to deploy DyG-Mamba in more real-world applications.

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

# A    Limitations

Although our DyG-Mamba delivers strong performance on dynamic graph modeling, it still has several notable limitations. ***(i) More Interactions***. We focus primarily on edge addition, which is widely studied interaction type in previous research. Extending our framework to other interaction types, such as node addition/deletion, edge deletion, and node/edge feature transformations, remain an avenue for future research. ***(ii) Larger-scale Datasets***. Existing benchmarks are relatively small-scale datasets. And it is unclear whether these findings will generalize to larger or more complex real-world scenarios. ***(iii) More Domains***. Although DyG-Mamba has achieved good results on dynamic network modeling, it is still unknown whether this conclusion can be extended to other domains. Recent insights from Mambaout [46] suggest that Mamba is especially suited for auto-regressive and long-sequence tasks. Even though dynamic link prediction and node classification are not strictly auto-regressive, we still obtain state-of-the-art performance, indicating that Mamba's broader effectiveness across diverse tasks warrants further exploration.

# B    Computational Complexity

In our implementation, we employ two DyG-Mamba layers with batch size $b$, feature dimension $d$, expanded state dimension $2d$, and SSM dimension $d_{ssm}$. Note that while Section 4.1 sets the feature dimension of node embeddings as $4d$, we use $d$ here for simplicity. On GPUs, high-bandwidth memory (HBM) provides larger capacity, whereas static random-access memory (SRAM) offers higher bandwidth. Building on Mamba, DyG-Mamba first reads $O(bL(2d) + (2d)d_{ssm})$ bytes of $(\mathbf{\Delta}, \boldsymbol{A}, \boldsymbol{B}, \boldsymbol{C})$ from slow HBM to fast SRAM. It then derives the discrete $\bar{\boldsymbol{A}}$, $\bar{\boldsymbol{B}}$ of size $(b, L, 2d, d_{ssm})$ in SRAM, executes the SSM operation in SRAM, and writes the output of size $(b, L, 2d)$ back to HBM. This approach reduces I/O overhead from $O(bL(2d)N)$ to $O(bL(2d) + (2d)d_{ssm})$, yielding a memory complexity of $O(bL(2d) + (2d)d_{ssm})$. Since $d_{ssm}$ is relatively smaller compared to $bL$, we simplify it as $O(bLd)$. The time complexity to calculate $\boldsymbol{B}, \boldsymbol{C}, \mathbf{\Delta}$ is $O(3bL(2d)d_{ssm})$, and the SSM process takes $O(bL(2d)d_{ssm})$. Compared to transformer-based methods with quadratic time and memory complexity $O(bL^2 d)$, DyG-Mamba scales effectively to large sequence lengths.

# C    Algorithm Details

Here, we list the detailed workflow of DyG-Mamba in Algorithm.1. We also list the procedures of DyG-Mamba for dynamic link prediction in Algorithm.2 and dynamic node classification in Algorithm.3.

---
**Algorithm 1** Continuous SSM

---
**Input**: Hidden states $\boldsymbol{Z} = \{\boldsymbol{z}_{i,1}, \boldsymbol{z}_{i,2}, \ldots, \boldsymbol{z}_{i,L}\}_{i=1}^{B}$: (B, L, $d$), Control signals $\mathbf{\Delta t} = \{\Delta t_1, \Delta t_2, \ldots, \Delta t_L\}$: (B, L, $d$), SSM dimension $d_{ssm}$, Expanded dimension: $2d$.

// Normalize the input sequence.

Initialize $\textbf{Parameter}_i^A$: $(2d, d_{ssm})$, $\textbf{Parameter}_i^\Delta$: $(d_{ssm})$

Initialize $\boldsymbol{x}$: (B, L, $2d$) $\leftarrow$ $\textbf{Linear}^{\boldsymbol{x}}(\boldsymbol{Z})$, $\boldsymbol{z}$: (B, L, $2d$) $\leftarrow$ $\textbf{Linear}^{\boldsymbol{z}}(\boldsymbol{Z})$, $\Delta t$: (B, L, $2d$) $\leftarrow$ $\textbf{Linear}^{\mathbf{\Delta t}}(\Delta t)$,

**for** *o in {forward, backward}* **do**

 $\boldsymbol{x}'_o$: (B, L, $2d$) $\leftarrow$ $\textbf{SiLU}(\textbf{Conv1d}_o(\boldsymbol{x}))$

 $\boldsymbol{B}_o$: (B, L, $d_{ssm}$) $\leftarrow$ $\textbf{Linear}_o^{B}(\boldsymbol{x}'_o)$

 $\boldsymbol{C}_o$: (B, L, $d_{ssm}$) $\leftarrow$ $\textbf{Linear}_o^{C}(\boldsymbol{x}'_o)$

 // Control signal to control the selection of historical information.

 $\Delta_o$: (B, L, $2d$) $\leftarrow$ Using Eq.(11)

 $\bar{\boldsymbol{A}}_o$: (B, L, $2d$, $d_{ssm}$) $\leftarrow$ $\Delta_i \otimes \textbf{Parameter}_o^A$

 $\bar{\boldsymbol{B}}_o$: (B, L, $2d$, $d_{ssm}$) $\leftarrow$ $\Delta_i \otimes \boldsymbol{B}_o$

 $\boldsymbol{y}_o$: (B, L, $2d$) $\leftarrow$ $\textbf{SSMs}(\bar{\boldsymbol{A}}_o, \bar{\boldsymbol{B}}_o, \boldsymbol{C}_o)(\boldsymbol{x}'_o)$

**end**

// Gated $\boldsymbol{y}$.

$\boldsymbol{y}'_{\text{forward}}$: (B, L, $2d$) $\leftarrow$ $\boldsymbol{y}_{\text{forward}} \odot \textbf{SiLU}(\boldsymbol{z})$

// Residual connection.

$\boldsymbol{y}'_{\text{backward}}$: (B, L, $2d$) $\leftarrow$ $\boldsymbol{y}_{\text{backward}} \odot \textbf{SiLU}(\boldsymbol{z})$

$\tilde{\boldsymbol{Z}}$: (B, L, $d$) $\leftarrow$ $\textbf{Linear}^T(\boldsymbol{y}'_{\text{forward}} + \boldsymbol{y}'_{\text{backward}}) + \boldsymbol{Z}$

**Output**: Updated hidden states $\tilde{\boldsymbol{Z}}$

---

**Algorithm 2** Dynamic Link Prediction

**Input:** The dynamic interaction set $\mathcal{D} = \{(u_i, v_i, t_i)\}_{i=1}^{K}$, Continuous SSM DyG-Mamba($\cdot$), readout function Read($\cdot$), output projection layer $\phi(\cdot)$.

**for** $T$ *Epochs* **do**
    **for** $(u_1, v_1, t) \in \mathcal{D}$ **do**
        For a node pair $(u_1, v_1)$. Sampled neighbor sequence: $S_1, S_2$.
        **for** $u$ *in* $\{u_1, v_1\}$ **do**
            Node Features: $\boldsymbol{X}_{u,N}^{\tau} \in \mathbb{R}^{|u| \times d_V}$,
            Edge Features: $\boldsymbol{X}_{u,E}^{\tau} \in \mathbb{R}^{|u| \times d_E}$,
            Time Features: $\boldsymbol{X}_{u,T}^{\tau} \in \mathbb{R}^{|u| \times d_T}$,
            Co-occurance Features: $\boldsymbol{X}_{u,C}^{\tau} \in \mathbb{R}^{|u| \times d_C}$,
            Time Span: $\boldsymbol{\Delta t} = \{\Delta t_1, \Delta t_2, \ldots, \Delta t_{|u|}\}$
            // `Feature Alignment.`
            $\boldsymbol{Z}_{u,*}^{\tau} = \boldsymbol{X}_{u,*}^{\tau} \boldsymbol{W}_* + \boldsymbol{b}$, where $* \in \{N, E, T, C\}$
            $\boldsymbol{Z}_u^{\tau} = \boldsymbol{Z}_{u,N}^{\tau} \| \boldsymbol{Z}_{u,E}^{\tau} \| \boldsymbol{Z}_{u,T}^{\tau} \| \boldsymbol{Z}_{u,C}^{\tau}$
            // `Continuous SSM encoder.`
            $\widehat{\boldsymbol{M}}_u^{\tau} = \text{DyG-Mamba}(\boldsymbol{Z}_u^{\tau}, \Delta t_u)$
        **end**
        $\widehat{\boldsymbol{Z}}_{u_1,\text{out}}^{\tau} = \phi(\text{Read}(\widehat{\boldsymbol{M}}_{u_1}^{\tau}))$, $\widehat{\boldsymbol{Z}}_{v_1,\text{out}}^{\tau} = \phi(\text{Read}(\widehat{\boldsymbol{M}}_{v_1}^{\tau}))$. // `Output.`
        $\hat{y} = \text{Softmax}(\text{Linear}(\text{RELU}(\text{Linear}(\widehat{\boldsymbol{Z}}_{u_1,\text{out}}^{\tau} \| \widehat{\boldsymbol{Z}}_{v_1,\text{out}}^{\tau}))))$.
    **end**
    Compute $\mathcal{L}_{\text{Link Prediction}}$.
**end**
**Output**: Dynamic link prediction labels.

---

**Algorithm 3** Dynamic Node Classification

**Input:** The dynamic interaction set $\mathcal{D} = \{(u_i, y_i)\}_{i=1}^{N}$, continuous SSM DyG-Mamba($\cdot$), readout function Read($\cdot$), output projection layer $\phi(\cdot)$.

**for** $T$ *Epochs* **do**
    **for** $(u, y) \in \mathcal{D}$ **do**
        Sampled neighbor sequence:
        $S = \{(k_1, t_1), (k_2, t_2), \ldots, (k_{|u|}, t_{|u|})\}$,
        Node Features: $\boldsymbol{X}_{u,N}^{\tau} \in \mathbb{R}^{|u| \times d_V}$,
        Edge Features: $\boldsymbol{X}_{u,E}^{\tau} \in \mathbb{R}^{|u| \times d_E}$,
        Time Features: $\boldsymbol{X}_{u,T}^{\tau} \in \mathbb{R}^{|u| \times d_T}$,
        Time Span: $\boldsymbol{\Delta t} = \{\Delta t_1, \Delta t_2, \ldots, \Delta t_L\}$
        $\boldsymbol{Z}_{u,*}^{\tau} = \boldsymbol{X}_{u,*}^{\tau} \boldsymbol{W}_* + \boldsymbol{b}$, where $* \in N, E, T$.
        $\boldsymbol{Z}_u^{\tau} = \boldsymbol{Z}_{u,N}^{\tau} \| \boldsymbol{Z}_{u,E}^{\tau} \| \boldsymbol{Z}_{u,T}^{\tau}$
        $\widehat{\boldsymbol{M}}_u^{\tau} = \text{DyG-Mamba}(\boldsymbol{Z}_u^{\tau}, \Delta t_u)$,
        $\widehat{\boldsymbol{Z}}_{u,\text{out}}^{\tau} = \phi(\text{Read}(\widehat{\boldsymbol{M}}_u^{\tau}))$,
        $\hat{y} = \text{Softmax}(\text{Linear}(\text{RELU}(\text{Linear}(\widehat{\boldsymbol{Z}}_{u,\text{out}}^{\tau}))))$.
    **end**
    Compute $\mathcal{L}_{\text{Node Classification}}$.
**end**
**Output:** Dynamic node classification labels.

---

Table 8: Statistics of the datasets. N/A denotes that there is no node/edge features. # Node denotes the number of nodes.

| Datasets | Domains | #Nodes | #Links | #N&L Feature | Bipartite | Duration | Unique Steps | Time Granularity |
|---|---|---|---|---|---|---|---|---|
| Wikipedia | Social | 9,227 | 157,474 | N/A & 172 | True | 1 month | 152,757 | Unix timestamps |
| Reddit | Social | 10,984 | 672,447 | N/A & 172 | True | 1 month | 669,065 | Unix timestamps |
| MOOC | Interaction | 7,144 | 411,749 | N/A & 4 | True | 17 months | 345,600 | Unix timestamps |
| LastFM | Interaction | 1,980 | 1,293,103 | N/A & N/A | True | 1 month | 1,283,614 | Unix timestamps |
| Enron | Social | 184 | 125,235 | N/A & N/A | False | 3 years | 22,632 | Unix timestamps |
| Social Evo. | Proximity | 74 | 2,099,519 | N/A & 2 | False | 8 months | 565,932 | Unix timestamps |
| UCI | Social | 1,899 | 59,835 | N/A & N/A | False | 196 days | 58,911 | Unix timestamps |
| Can. Parl. | Politics | 734 | 74,478 | N/A & 1 | False | 14 years | 14 | years |
| US Legis. | Politics | 225 | 60,396 | N/A & 1 | False | 12 congresses | 12 | congresses |
| UN Trade | Economics | 255 | 507,497 | N/A & 1 | False | 32 years | 32 | years |
| UN Vote | Politics | 201 | 1,035,742 | N/A & 1 | False | 72 years | 72 | years |
| Contact | Proximity | 692 | 2,426,279 | N/A & 1 | False | 1 month | 8,064 | 5 minutes |

# D Experimental Details

## D.1 Dataset Details

We evaluate our methods on a diverse set of dynamic graph datasets, including twelve publicly available datasets collected by Edgebank [30], which are publicly available[1]. We present the statistics of the datasets in Table 8, where #N&L Feature stands for the dimensions of the node and link features. Note that our calculation of the Contact dataset's statistics (694 nodes and 2,426,280 links) slightly differs from the values reported in [30], although both are derived from the same dataset.

Table 9: Configurations

| Configuration | Setting |
|---|---|
| Learning rate | 0.0001 |
| Train Epochs | 100 |
| Optimizer | Adam |
| Dimension of time encoding $d_T$ | 100 |
| Dimension of co-occurrence $d_C$ | 50 |
| Dimension of aligned encoding $d$ | 50 |
| Dimension of $\Delta t_i$'s encoder $4d$ | 200 |
| Dimension of output $d_{out}$ | 172 |
| Number of Mamba blocks | 2 |
| Dimension of SSM $d_{ssm}$ | 16 |
| Expanded factor of Mamba | 2 |
| Number of Corss-Attention layer | 1 |

Table 10: Sequence Length Settings

| Dataset | Sequence Length |
|---|---|
| Wikipedia | 64 |
| Reddit | 64 |
| MOOC | 256 |
| LastFM | 512 |
| Enron | 512 |
| Social Evo. | 64 |
| UCI | 32 |
| Can. Parl. | 2048 |
| US Legis. | 272 |
| UN Trade | 256 |
| UN Vote | 128 |
| Contact | 32 |

## D.2 Implementation Details

**Experiment Environment**. We conduct experiments on an Ubuntu 22.04 LTS server equipped with one Intel(R) Core(TM) i9-10900X CPU @ 3.70GHz with 10 physical cores and NVIDIA RTX A6000 GPUs (48GB). The code is written in Python 3.10 and we use PyTorch 2.1.0 on CUDA 11.8 to train the model.

**Configuration Details**. For all baselines, we follow the configurations as DyGFormer reported [6]. For DyG-Mamba, we list all configurations in Table 9. Then, we perform the grid search to find the optimal sequence length, with a search range spanning from $32$ to $2,048$ in powers of 2. It is worth noticing that DyG-Mamba can handle nodes with sequence lengths shorter than the defined length. When the sequence length exceeds the specified length, we truncate the sequence and preserve the most recent interactions up to the defined length. Finally, we present the sequence length settings in Table 10. All implementation details could be accessed at the link: https://anonymous.4open/DyGMamba.

---

[1] https://zenodo.org/records/dynamic-graphs

**The inconsistency problem between the description of the transductive setting in DyGFormer and the coding in DyGLib.** Thanks to other researchers in this community, we observed that CAW-N explicitly avoids including unseen nodes in the validate/test sets. In contrast, DyGLib, TGAT, and TGN adopt a more relaxed version of the transductive setting, where both previously observed and new nodes can appear during evaluation. For a fair comparison, we follow the coding in DyGLib, which uses more relaxed version of the transductive setting. Therefore all baselines in our paper use the same data split with both previously observed and new nodes in transductive setting, to ensure the fairness of experimental comparisons. To compare with CAW-N, we also re-implement CAW-N with the relaxed version of transductive setting.

## D.3 Detailed Evaluation Settings

**Detailed Settings for Effectiveness Evaluation.** To provide a more comprehensive evaluation of baseline performance, and following [30], we adopt three distinct negative edge sampling strategies for the temporal link prediction task. We define the training and test edge sets as $E_{\text{train}}$ and $E_{\text{test}}$, respectively. The edges of a given dynamic graph can then be grouped into three categories: (a) edges observed only during training ($E_{\text{train}} \setminus E_{\text{test}}$), (b) edges that appear both in training and test ($E_{\text{train}} \cap E_{\text{test}}$), referred to as *transductive* edges, and (c) edges observed exclusively in the test phase ($E_{\text{test}} \setminus E_{\text{train}}$), regarded as *inductive* edges. Note that inductive negative sampling here is distinct from the usual notion of inductive settings. We elaborate on the three sampling strategies as follows:

- **Random Negative Sampling (*rnd*).** Negative edges are sampled at random from all possible node pairs. At each timestep, we retain the timestamps, features, and source nodes of the positive edges, but randomly select their destination nodes from the entire node set.

- **Historical Negative Sampling (*hist*).** In historical negative sampling, we focus on edges that were observed at previous time steps but are absent in the current step. This approach assesses whether a method can accurately predict the specific timestamps at which an edge may reappear, rather than simply predicting that it always reoccurs once observed. Formally, for a given time step $t$, we sample from edges in $(E_{train} \cap \overline{E_t})$. If the number of available historical edges is insufficient to match the number of positive edges, we revert to random sampling for the remainder.

- **Inductive Negative Sampling (*ind*).** Whereas historical sampling centers on edges observed during training, inductive negative sampling evaluates whether a model can capture the reoccurrence of edges that first appear only at test time. Once newly appearing edges have been observed in the test, the model is asked to predict if these edges will reoccur in subsequent time steps. Formally, at time $t$, we sample from $(E_{test} \cap \overline{E_{train}} \cap \overline{E_t})$. If there are not enough such inductive edges to match the number of positive edges, the remaining negative edges are sampled randomly.

**Detailed Settings for Efficiency Evaluation.** In Figures 3,4, in the evaluation of time and memory consumption, we do not choose the configuration as reported in DyGLib[1], as there is a significant difference in sequence lengths between different baselines. In the LastFM dataset, the reported number of sampled neighbors for DyRep, TGN, and GraphMixer is set to 10, while for DyGFormer and DyG-Mamba, it is 512. This is unfair because the complexity of time and memory is highly dependent on the number of neighbors sampled. Therefore, for a fair comparison in terms of time and memory consumption, we use the same number of sampled neighbors across all models: 32 for UCI, 256 for Enron, 512 for LastFM and 64 for Reddit.

**Detailed Settings for Robustness Evaluation.** As shown in Figure 5, the purpose of the robustness test is to evaluate the ability to against noise in edges and timestamps. In the training step, we typically train models on a transductive setting with random negative sampling. In the evaluation step, after neighbor sampling, we randomly select $\sigma * L$ positions to insert noise. Specifically, the noise position's node and timestamps are randomly generated. And the noise rate $\sigma$ is chosen from 0.1 to 0.6.

---

[1] github.com/yule-BUAA/DyGLib

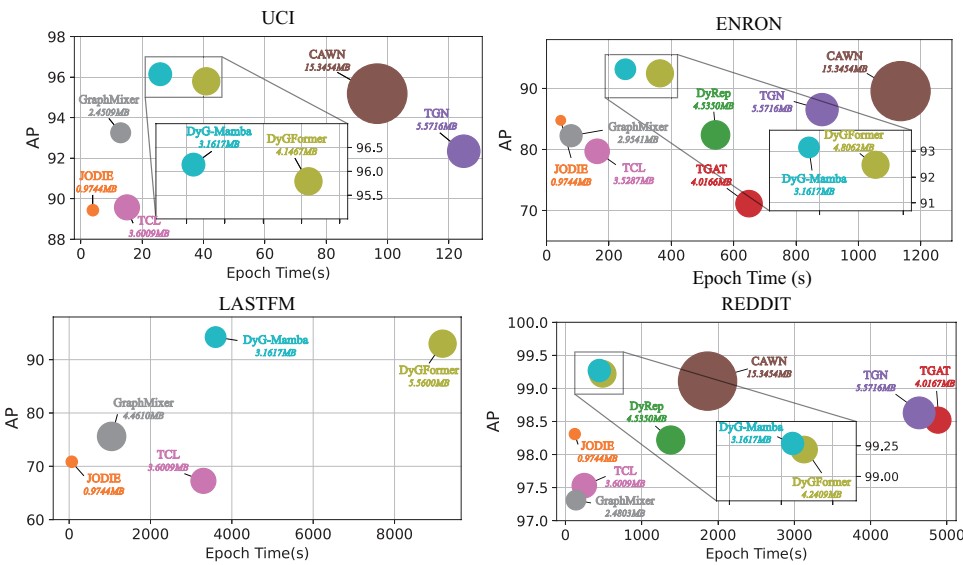

Figure 7: The AP score with different sequence lengths. We use the same sequence length for each model (uci=32, enron=256, lastfm=512, reddit=64), The not appearing model means OOM.

## D.4 Additional Experimental Results

**Training Time and Parameter Size.** Figure 7 shows the additional time and parameter size comparison on the UCI, Enron, LastFM, and Reddit datasets. The models that do not appear in the figure experienced out-of-memory (OOM) errors. We can see that our model achieves the best performance while incurring low time and memory costs.

Table 11: Performance on dynamic node classification.

| Methods | Wikipedia | Reddit | Avg. Rank |
|---|---|---|---|
| JODIE | **88.99±1.05** | 60.37±2.58 | 5.00 |
| DyRep | 86.39±0.98 | 63.72±1.32 | 6.00 |
| TGAT | 84.09±1.27 | 70.04±1.09 | 5.00 |
| TGN | 86.38±2.34 | 63.27±0.90 | 7.00 |
| CAWN | 84.88±1.33 | 66.34±1.78 | 6.00 |
| EdgeBank | N/A | N/A | N/A |
| TCL | 77.83±2.13 | 68.87±2.15 | 6.00 |
| GraphMixer | 86.80±0.79 | 64.22±3.32 | 5.00 |
| DyGFormer | 87.44±1.08 | 68.00±1.74 | 3.50 |
| DyG-Mamba | 88.58±0.92 | **70.79±1.97** | **1.50** |

**Performance on Dynamic Node Classification.** For dynamic node classification, we estimate the state of a node in a given interaction at a specific time and use *AUC-ROC* as the evaluation metric. Table 11 shows the AUC-ROC results on dynamic node classification. DyG-Mamba achieves SOTA performance on the Reddit dataset and second-best performance on the Wikipedia dataset. In addition, DyG-Mamba achieves the best average rank of 1.5 compared to the second-best DyGFormer with 3.5 AUC-ROC results for all baselines in Table 11.

**Robustness Evaluation.** We provide additional robustness test results on the UCI and Can. Parl. datasets, as shown in Figure 8. DyG-Mamba consistently exhibits greater robustness compared to DyGFormer and GraphMixer. However, TGN uses a memory bank to store previous node embeddings, making it less sensitive to noise in current neighbors.

**Transductive Dynamic Link Prediction.** We show the AUC-ROC for transductive dynamic link prediction with three negative sampling strategies in Table 15. Since we cannot reproduce the same performance reported by FreeDyG [10], we directly copy the results from their paper.

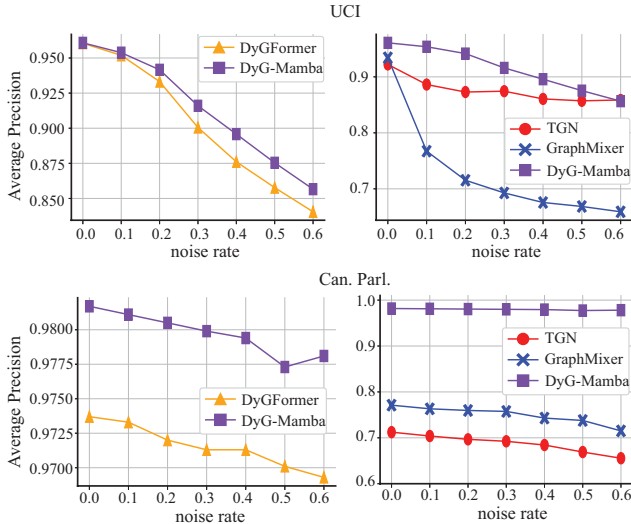

Figure 8: The robustness test on datasets UCI, Can. Parl.

**Inductive Dynamic Link Prediction.** We present the AP and AUC-ROC for inductive dynamic link prediction with three negative sampling strategies in Table 16 and Table 17.

**Comparison of the effectiveness of Co-occurrence.** DyGFormer designs co-occurrence module and has conducted ablation studies on it. DyG-Mamba focuses on detailed ablations of the modules we propose. But we also conduct additional ablation to compare the effectiveness of co-occurrence encoding in table 12. From the results, DyG-Mamba still exhibits a strong ability to capture long-term dependencies, outperforming DyGFormer.

Table 12: Comparison of the effectiveness of Co-occurrence Frequency Encoding.

|  | uci | USLegis | UN Trade |
| --- | --- | --- | --- |
| DyG-Mamba | 96.79±0.08 | 74.11±2.32 | 68.55±0.16 |
| DyG-Mamba w/o Co-occurrence | 93.09±1.31 | 73.53±2.43 | 66.32±0.55 |
| DyGFormer | 95.79±0.17 | 71.11±0.59 | 66.46±1.29 |
| DyGFormer w/o Co-occurrence | 83.05±0.38 | 70.59±0.36 | 61.93±1.79 |

**Ablation Study with Co-occurrence and skip connection.** We conduct ablation studies on three datasets. 'w/o Co-occurrence' refers to removing Co-occurrence Frequency Encoding from DyG-Mamba, while 'w/o skip connection' denotes the removal of the skip connection in the SSM.

Table 13: We conduct ablation studies on three datasets. 'w/o Co-occurrence' refers to removing Co-occurrence Frequency Encoding from DyG-Mamba, while 'w/o skip connection' denotes the removal of the skip connection in the SSM.

|  | UCI | US Legis | UN Trade |
| --- | --- | --- | --- |
| DyG-Mamba | 96.79 ± 0.08 | 74.11 ± 2.32 | 68.55 ± 0.16 |
| w/o Co-occurrence | 93.09 ± 1.31 | 73.53 ± 2.43 | 66.32 ± 0.55 |
| w/o skip-connection | 95.37 ± 0.06 | 73.64 ± 2.51 | 67.64 ± 0.25 |

**Performance with different sequence length.** As shown in Table 14, DyG-Mamba can achieve better performance with longer input sequences. However, to ensure a fair comparison, we follow the same input sequence length as DyGFormer rather than incorporating additional historical information.

**Hyperparameter Sensitivity.** DyG-Mamba is insensitive to hyperparameters, see Figure 9. Therefore, the hyperparameters listed in Table 6 are applied consistently across all experiments.

Table 14: AP score across different sequence length. '*' denotes the sequence length used in DyGFormer.

|  | 32 | 256 | 512 | 1024 | 2048 | 4096 |
|---|---|---|---|---|---|---|
| **uci** | 96.79±0.08* | 96.82±0.09 | 96.95±0.06 | 97.38±0.05 | 97.45±0.03 | 97.47±0.08 |
| **USLegis** | 73.99±1.52 | 74.11±2.32* | 74.17±2.14 | 74.34±2.17 | 74.47±2.44 | 74.96±2.04 |

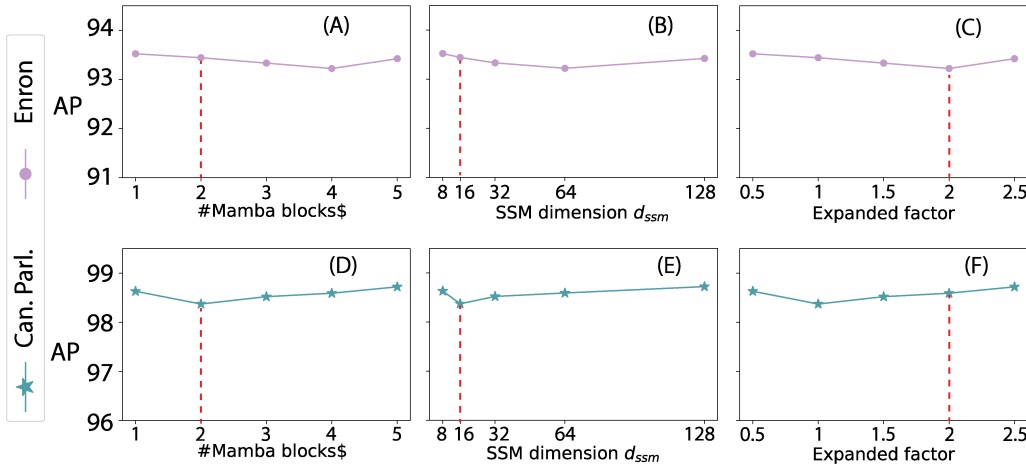

Figure 9: We tune the parameters of DyG-Mamba, including the number of blocks, the SSM dimension, and the expansion factor, on two datasets: Enron (A-C), Can.Parl. (D-F). For each parameter, we adjust its value while keeping the others fixed at DyG-Mamba's default setting (the red line).

Table 15: AUC-ROC for transductive dynamic link prediction with random, historical, and inductive negative sampling strategies.

|  | Datasets | JODIE | DyRep | TGAT | TGN | CAWN | EdgeBank | TCL | GraphMixer | FreeDyG | DyGFormer | DyG-Mamba |
|---|---|---|---|---|---|---|---|---|---|---|---|---|
| rnd | Wikipedia | 96.33 ± 0.07 | 94.37 ± 0.09 | 96.67 ± 0.07 | 98.37 ± 0.07 | 98.54 ± 0.04 | 90.78 ± 0.00 | 95.84 ± 0.18 | 96.92 ± 0.03 | 99.41 ± 0.01 | 98.91 ± 0.02 | 98.96 ± 0.00 |
|  | Reddit | 98.31 ± 0.05 | 98.17 ± 0.05 | 98.47 ± 0.02 | 98.60 ± 0.06 | 99.01 ± 0.01 | 95.37 ± 0.00 | 97.42 ± 0.02 | 97.17 ± 0.02 | 99.50 ± 0.01 | 99.15 ± 0.01 | 99.20 ± 0.00 |
|  | MOOC | 83.81 ± 2.09 | 85.03 ± 0.58 | 87.11 ± 0.19 | 91.21 ± 1.15 | 80.38 ± 0.26 | 60.86 ± 0.00 | 83.12 ± 0.18 | 84.01 ± 0.17 | 89.93 ± 0.35 | 87.91 ± 0.58 | 90.93 ± 0.13 |
|  | LastFM | 70.49 ± 1.66 | 71.16 ± 1.89 | 71.59 ± 0.18 | 78.47 ± 2.94 | 85.92 ± 0.10 | 83.77 ± 0.00 | 64.06 ± 1.16 | 73.53 ± 0.12 | 93.42 ± 0.15 | 93.05 ± 0.10 | 93.99 ± 0.02 |
|  | Enron | 87.96 ± 0.52 | 84.89 ± 3.00 | 68.89 ± 1.10 | 88.32 ± 0.99 | 90.45 ± 0.14 | 87.05 ± 0.00 | 75.74 ± 0.72 | 84.38 ± 0.21 | 94.01 ± 0.11 | 93.33 ± 0.13 | 93.03 ± 0.06 |
|  | Social Evo. | 92.05 ± 0.46 | 90.76 ± 0.21 | 94.76 ± 0.16 | 95.39 ± 0.17 | 87.34 ± 0.08 | 81.60 ± 0.00 | 94.84 ± 0.17 | 95.23 ± 0.07 | 96.59 ± 0.04 | 96.30 ± 0.01 | 96.39 ± 0.01 |
|  | UCI | 90.44 ± 0.49 | 68.77 ± 2.34 | 78.53 ± 0.74 | 92.03 ± 1.13 | 93.87 ± 0.08 | 77.30 ± 0.00 | 87.82 ± 1.36 | 91.81 ± 0.67 | 95.00 ± 0.21 | 94.49 ± 0.26 | 96.50 ± 0.06 |
|  | Can. Parl. | 78.21 ± 0.23 | 73.35 ± 3.67 | 75.69 ± 0.78 | 76.99 ± 1.80 | 75.70 ± 3.27 | 64.14 ± 0.00 | 72.46 ± 3.23 | 83.17 ± 0.53 | N/A | 97.76 ± 0.41 | 98.77 ± 0.05 |
|  | US Legis. | 82.85 ± 1.07 | 82.28 ± 0.32 | 75.84 ± 1.99 | 83.34 ± 0.43 | 77.16 ± 0.39 | 62.57 ± 0.00 | 76.27 ± 0.63 | 76.96 ± 0.79 | N/A | 77.90 ± 0.58 | 78.27 ± 2.80 |
|  | UN Trade | 69.62 ± 0.44 | 67.44 ± 0.83 | 64.01 ± 0.12 | 69.10 ± 1.67 | 68.54 ± 0.18 | 66.75 ± 0.00 | 64.72 ± 0.05 | 65.52 ± 0.51 | N/A | 70.20 ± 1.44 | 72.25 ± 0.07 |
|  | UN Vote | 68.53 ± 0.95 | 67.18 ± 1.04 | 52.83 ± 1.12 | 69.71 ± 2.65 | 53.09 ± 0.22 | 62.97 ± 0.00 | 51.88 ± 0.36 | 52.46 ± 0.27 | N/A | 57.12 ± 0.62 | 69.58 ± 0.55 |
|  | Contact | 96.66 ± 0.89 | 96.48 ± 0.14 | 96.95 ± 0.08 | 97.54 ± 0.35 | 89.99 ± 0.34 | 94.34 ± 0.00 | 94.15 ± 0.09 | 93.94 ± 0.02 | N/A | 98.53 ± 0.01 | 98.58 ± 0.01 |
|  | Avg. Rank | 5.33 | 6.75 | 7.00 | 3.25 | 5.58 | 8.17 | 8.25 | 6.58 | N/A | 2.58 | 1.50 |
| hist | Wikipedia | 80.77 ± 0.73 | 77.74 ± 0.33 | 82.87 ± 0.22 | 82.74 ± 0.32 | 67.84 ± 0.64 | 77.27 ± 0.00 | 85.76 ± 0.46 | 87.68 ± 0.17 | 82.78 ± 0.30 | 78.80 ± 1.95 | 78.93 ± 1.42 |
|  | Reddit | 80.52 ± 0.32 | 80.15 ± 0.18 | 79.33 ± 0.16 | 81.11 ± 0.19 | 80.27 ± 0.30 | 78.58 ± 0.00 | 76.49 ± 0.16 | 77.80 ± 0.12 | 85.92 ± 0.10 | 80.54 ± 0.29 | 80.96 ± 0.20 |
|  | MOOC | 82.75 ± 0.83 | 81.06 ± 0.94 | 80.81 ± 0.67 | 88.00 ± 1.80 | 71.57 ± 1.07 | 61.90 ± 0.00 | 72.09 ± 0.56 | 76.68 ± 1.40 | 88.32 ± 0.99 | 87.04 ± 0.35 | 88.74 ± 0.97 |
|  | LastFM | 75.22 ± 2.36 | 74.65 ± 1.98 | 64.27 ± 0.26 | 77.97 ± 3.04 | 77.88 ± 0.24 | 78.09 ± 0.00 | 47.24 ± 3.13 | 64.21 ± 0.73 | 73.53 ± 0.12 | 78.78 ± 0.35 | 80.88 ± 0.52 |
|  | Enron | 75.39 ± 2.37 | 74.69 ± 3.55 | 61.85 ± 1.43 | 77.09 ± 2.22 | 65.10 ± 0.34 | 79.59 ± 0.00 | 67.95 ± 0.88 | 75.27 ± 1.14 | 75.74 ± 0.72 | 76.55 ± 0.52 | 78.09 ± 0.65 |
|  | Social Evo. | 90.06 ± 3.15 | 93.12 ± 0.34 | 93.08 ± 0.59 | 94.71 ± 0.53 | 87.43 ± 0.15 | 85.81 ± 0.00 | 93.44 ± 0.68 | 94.39 ± 0.31 | 97.42 ± 0.02 | 97.28 ± 0.07 | 96.58 ± 0.24 |
|  | UCI | 78.64 ± 3.50 | 57.91 ± 3.12 | 58.89 ± 1.57 | 77.25 ± 2.68 | 57.86 ± 0.15 | 69.56 ± 0.00 | 72.25 ± 3.46 | 77.54 ± 2.02 | 80.38 ± 0.26 | 76.97 ± 0.24 | 77.35 ± 1.25 |
|  | Can. Parl. | 62.44 ± 1.11 | 70.16 ± 1.70 | 70.86 ± 0.94 | 73.23 ± 3.08 | 72.06 ± 3.94 | 63.04 ± 0.00 | 69.95 ± 3.70 | 79.03 ± 1.01 | N/A | 97.61 ± 0.40 | 96.97 ± 0.30 |
|  | US Legis. | 67.47 ± 6.40 | 91.44 ± 1.18 | 73.47 ± 5.25 | 83.53 ± 4.53 | 78.62 ± 7.46 | 67.41 ± 0.00 | 83.97 ± 3.71 | 85.17 ± 0.70 | N/A | 90.77 ± 1.96 | 97.11±0.28 |
|  | UN Trade | 68.92 ± 1.40 | 64.36 ± 1.40 | 60.37 ± 0.68 | 63.93 ± 5.41 | 63.09 ± 0.74 | 86.61 ± 0.00 | 61.43 ± 1.04 | 63.20 ± 1.54 | N/A | 73.86 ± 1.13 | 75.24±0.16 |
|  | UN Vote | 76.84 ± 1.01 | 74.72 ± 1.43 | 53.95 ± 3.15 | 73.40 ± 5.20 | 51.27 ± 0.33 | 89.62 ± 0.00 | 52.29 ± 2.39 | 52.61 ± 1.44 | N/A | 64.27 ± 1.78 | 64.87±4.51 |
|  | Contact | 96.35 ± 0.92 | 96.00 ± 0.23 | 95.39 ± 0.43 | 93.76 ± 1.29 | 83.06 ± 0.22 | 92.17 ± 0.00 | 93.34 ± 0.19 | 93.14 ± 0.34 | N/A | 97.17 ± 0.05 | 97.43 ± 0.11 |
|  | Avg. Rank | 5.00 | 5.67 | 7.08 | 3.92 | 8.25 | 6.50 | 7.25 | 5.67 | N/A | 3.33 | 2.33 |
| ind | Wikipedia | 70.96 ± 0.78 | 67.36 ± 0.96 | 81.93 ± 0.22 | 80.97 ± 0.31 | 70.95 ± 0.95 | 81.73 ± 0.00 | 82.19 ± 0.48 | 84.28 ± 0.30 | 82.74 ± 0.32 | 75.09 ± 3.70 | 78.69 ± 2.23 |
|  | Reddit | 83.51 ± 0.15 | 82.90 ± 0.31 | 87.13 ± 0.20 | 84.56 ± 0.24 | 88.04 ± 0.29 | 85.93 ± 0.00 | 84.67 ± 0.29 | 82.21 ± 0.13 | 84.38 ± 0.21 | 86.23 ± 0.51 | 87.22 ± 0.52 |
|  | MOOC | 66.63 ± 2.30 | 63.26 ± 1.01 | 73.18 ± 0.33 | 77.44 ± 2.86 | 70.32 ± 1.43 | 48.18 ± 0.00 | 70.36 ± 0.37 | 72.45 ± 0.72 | 78.47 ± 0.94 | 80.76 ± 0.76 | 82.02 ± 1.22 |
|  | LastFM | 61.32 ± 3.49 | 62.15 ± 2.12 | 63.99 ± 0.21 | 65.46 ± 4.27 | 67.92 ± 0.44 | 77.37 ± 0.00 | 46.93 ± 2.59 | 60.22 ± 0.32 | 72.30 ± 0.59 | 69.25 ± 0.36 | 68.83 ± 0.53 |
|  | Enron | 70.92 ± 1.05 | 68.73 ± 1.34 | 60.45 ± 2.12 | 71.34 ± 2.46 | 75.17 ± 0.50 | 75.00 ± 0.00 | 67.64 ± 0.86 | 71.53 ± 0.85 | 77.27 ± 0.61 | 74.07 ± 0.64 | 77.79 ± 0.54 |
|  | Social Evo. | 90.01 ± 3.19 | 93.07 ± 0.38 | 92.94 ± 0.61 | 95.24 ± 0.56 | 89.93 ± 0.15 | 87.88 ± 0.00 | 93.44 ± 0.72 | 94.22 ± 0.32 | 98.47 ± 0.02 | 97.51 ± 0.06 | 96.78 ± 0.21 |
|  | UCI | 64.14 ± 1.26 | 54.25 ± 2.01 | 60.80 ± 1.01 | 64.11 ± 1.04 | 58.06 ± 0.26 | 58.03 ± 0.00 | 70.05 ± 1.86 | 74.59 ± 0.74 | 75.39 ± 0.57 | 65.96 ± 1.18 | 67.36 ± 2.58 |
|  | Can. Parl. | 52.88 ± 0.80 | 63.53 ± 0.65 | 72.47 ± 1.18 | 69.57 ± 2.81 | 72.93 ± 1.78 | 61.41 ± 0.00 | 69.47 ± 2.12 | 70.52 ± 0.94 | N/A | 96.70 ± 0.59 | 96.99 ± 0.65 |
|  | US Legis. | 59.05 ± 5.52 | 89.44 ± 0.71 | 71.62 ± 5.42 | 78.12 ± 4.46 | 76.45 ± 7.02 | 68.66 ± 0.00 | 82.54 ± 3.91 | 84.22 ± 0.91 | N/A | 87.96 ± 1.80 | 90.51 ± 0.26 |
|  | UN Trade | 66.82 ± 1.27 | 65.60 ± 1.28 | 66.13 ± 0.78 | 66.37 ± 5.39 | 71.73 ± 0.74 | 74.20 ± 0.00 | 67.80 ± 1.21 | 66.53 ± 1.22 | N/A | 62.56 ± 1.51 | 70.82 ± 1.14 |
|  | UN Vote | 73.73 ± 1.61 | 72.80 ± 2.16 | 53.04 ± 2.58 | 72.69 ± 3.72 | 52.75 ± 0.90 | 72.85 ± 0.00 | 52.02 ± 1.64 | 51.89 ± 0.74 | N/A | 53.37 ± 1.26 | 62.39 ± 2.21 |
|  | Contact | 94.47 ± 1.08 | 94.23 ± 0.18 | 94.10 ± 0.41 | 91.64 ± 1.72 | 87.68 ± 0.24 | 85.87 ± 0.00 | 91.23 ± 0.19 | 90.96 ± 0.27 | N/A | 95.01 ± 0.15 | 94.94 ± 0.21 |
|  | Avg. Rank | 6.75 | 7.08 | 6.00 | 5.33 | 5.75 | 6.08 | 6.00 | 5.67 | N/A | 3.83 | 2.50 |

Table 16: AP for inductive dynamic link prediction with random, historical, and inductive negative sampling strategies.

| | Datasets | JODIE | DyRep | TGAT | TGN | CAWN | TCL | GraphMixer | FreeDyG | DyGFormer | DyG-Mamba |
|---|---|---|---|---|---|---|---|---|---|---|---|
| rnd | Wikipedia | 94.82 ± 0.20 | 92.43 ± 0.37 | 96.22 ± 0.07 | 97.83 ± 0.04 | 98.24 ± 0.03 | 96.22 ± 0.17 | 96.65 ± 0.02 | 98.97 ± 0.01 | 98.59 ± 0.03 | 98.66 ± 0.02 |
| | Reddit | 96.50 ± 0.13 | 96.09 ± 0.11 | 97.09 ± 0.04 | 97.50 ± 0.07 | 98.62 ± 0.01 | 94.09 ± 0.07 | 95.26 ± 0.02 | 98.91 ± 0.01 | 98.84 ± 0.02 | 98.91 ± 0.01 |
| | MOOC | 79.63 ± 1.92 | 81.07 ± 0.44 | 85.50 ± 0.19 | 89.04 ± 1.17 | 81.42 ± 0.24 | 80.60 ± 0.22 | 81.41 ± 0.21 | 87.75 ± 0.62 | 86.96 ± 0.43 | 89.98 ± 0.46 |
| | LastFM | 81.61 ± 3.82 | 83.02 ± 1.48 | 78.63 ± 0.31 | 81.45 ± 4.29 | 89.42 ± 0.07 | 73.53 ± 1.66 | 82.11 ± 0.42 | 94.89 ± 0.01 | 94.23 ± 0.09 | 95.16 ± 0.05 |
| | Enron | 80.72 ± 1.39 | 74.55 ± 3.95 | 67.05 ± 1.51 | 77.94 ± 1.02 | 86.35 ± 0.51 | 76.14 ± 0.79 | 75.88 ± 0.48 | 89.69 ± 0.17 | 89.76 ± 0.34 | 90.97 ± 0.01 |
| | Social Evo. | 91.96 ± 0.48 | 90.04 ± 0.47 | 91.41 ± 0.16 | 90.77 ± 0.86 | 79.94 ± 0.18 | 91.55 ± 0.09 | 91.86 ± 0.06 | 94.76 ± 0.05 | 93.14 ± 0.04 | 93.17 ± 0.05 |
| | UCI | 79.86 ± 1.48 | 57.48 ± 1.87 | 79.54 ± 0.48 | 88.12 ± 2.05 | 92.73 ± 0.06 | 87.36 ± 2.03 | 91.19 ± 0.42 | 94.85 ± 0.10 | 94.54 ± 0.12 | 93.38 ± 0.21 |
| | Can. Parl. | 53.92 ± 0.94 | 54.02 ± 0.76 | 55.18 ± 0.79 | 54.10 ± 0.93 | 55.80 ± 0.69 | 54.30 ± 0.66 | 55.91 ± 0.82 | N/A | 87.74 ± 0.71 | 96.64 ± 0.10 |
| | US Legis. | 54.93 ± 2.29 | 57.28 ± 0.71 | 51.00 ± 3.11 | 58.63 ± 0.37 | 53.17 ± 1.20 | 52.59 ± 0.97 | 50.71 ± 0.76 | N/A | 54.28 ± 2.87 | 55.25 ± 4.95 |
| | UN Trade | 59.65 ± 0.77 | 57.02 ± 0.69 | 61.03 ± 0.18 | 58.31 ± 3.15 | 65.24 ± 0.21 | 62.21 ± 0.12 | 62.17 ± 0.31 | N/A | 64.55 ± 0.62 | 67.04 ± 0.20 |
| | UN Vote | 56.64 ± 0.96 | 54.62 ± 2.22 | 52.24 ± 1.46 | 58.85 ± 2.51 | 49.94 ± 0.45 | 51.60 ± 0.97 | 50.68 ± 0.44 | N/A | 55.93 ± 0.39 | 58.08 ± 0.55 |
| | Contact | 94.34 ± 1.45 | 92.18 ± 0.41 | 95.87 ± 0.11 | 93.82 ± 0.99 | 89.55 ± 0.30 | 91.11 ± 0.12 | 90.59 ± 0.05 | N/A | 98.03 ± 0.02 | 98.10 ± 0.02 |
| | Avg. Rank | 5.83 | 6.83 | 6.21 | 4.67 | 4.92 | 6.71 | 6.00 | N/A | 2.50 | 1.33 |
| hist | Wikipedia | 68.69 ± 0.39 | 62.18 ± 1.27 | 84.17 ± 0.22 | 81.76 ± 0.32 | 67.27 ± 1.63 | 82.20 ± 2.18 | 87.60 ± 0.30 | 82.78 ± 0.30 | 71.42 ± 4.43 | 73.68 ± 4.06 |
| | Reddit | 62.34 ± 0.54 | 61.60 ± 0.72 | 63.47 ± 0.36 | 64.85 ± 0.85 | 63.67 ± 0.41 | 60.83 ± 0.25 | 64.50 ± 0.26 | 66.02 ± 0.41 | 65.37 ± 0.60 | 66.74 ± 0.13 |
| | MOOC | 63.22 ± 1.55 | 62.93 ± 1.24 | 76.73 ± 0.29 | 77.07 ± 3.41 | 74.68 ± 0.68 | 74.27 ± 0.53 | 74.00 ± 0.97 | 81.63 ± 0.33 | 80.82 ± 0.30 | 81.64 ± 0.67 |
| | LastFM | 70.39 ± 4.31 | 71.45 ± 1.76 | 76.27 ± 0.25 | 66.65 ± 6.11 | 71.33 ± 0.47 | 65.78 ± 0.65 | 76.42 ± 0.22 | 77.28 ± 0.21 | 76.35 ± 0.52 | 79.22 ± 0.33 |
| | Enron | 65.86 ± 3.71 | 62.08 ± 2.27 | 61.40 ± 1.31 | 62.91 ± 1.16 | 60.70 ± 0.36 | 67.11 ± 0.62 | 72.37 ± 1.37 | 73.01 ± 0.88 | 67.07 ± 0.62 | 75.12 ± 1.43 |
| | Social Evo. | 88.51 ± 0.87 | 88.72 ± 1.10 | 93.97 ± 0.54 | 90.66 ± 1.62 | 79.83 ± 0.39 | 94.10 ± 0.31 | 94.01 ± 0.47 | 96.82 ± 0.16 | | 95.45 ± 0.30 |
| | UCI | 63.11 ± 2.27 | 52.47 ± 2.06 | 70.52 ± 0.93 | 70.78 ± 0.78 | 64.54 ± 0.47 | 76.71 ± 1.00 | 81.66 ± 0.49 | 82.35 ± 0.39 | 72.13 ± 1.87 | 73.65 ± 3.70 |
| | Can. Parl. | 52.60 ± 0.88 | 52.28 ± 0.31 | 56.72 ± 0.47 | 54.42 ± 0.77 | 57.14 ± 0.07 | 55.71 ± 0.74 | 55.84 ± 0.73 | N/A | 87.40 ± 0.85 | 91.59 ± 1.10 |
| | US Legis. | 52.94 ± 2.11 | 62.10 ± 1.41 | 51.83 ± 3.95 | 61.18 ± 1.10 | 55.56 ± 1.71 | 53.87 ± 1.41 | 52.03 ± 1.02 | N/A | 56.31 ± 3.46 | 55.90 ± 0.74 |
| | UN Trade | 55.46 ± 1.19 | 55.49 ± 0.84 | 55.28 ± 0.71 | 52.80 ± 3.19 | 55.00 ± 0.38 | 55.76 ± 1.03 | 54.94 ± 0.97 | N/A | 53.20 ± 1.07 | 56.80 ± 0.27 |
| | UN Vote | 61.04 ± 1.30 | 60.22 ± 1.78 | 53.05 ± 3.10 | 63.74 ± 3.00 | 47.98 ± 0.84 | 54.19 ± 2.17 | 48.09 ± 0.43 | N/A | 52.63 ± 1.26 | 58.46 ± 0.91 |
| | Contact | 90.42 ± 2.34 | 89.22 ± 0.66 | 94.15 ± 0.45 | 88.13 ± 1.50 | 74.20 ± 0.80 | 90.44 ± 0.17 | 89.91 ± 0.36 | N/A | 93.56 ± 0.52 | 93.66 ± 0.56 |
| | Avg. Rank | 6.33 | 6.42 | 5.00 | 5.17 | 6.75 | 4.83 | 4.58 | N/A | 3.75 | 2.17 |
| ind | Wikipedia | 68.70 ± 0.39 | 62.19 ± 1.28 | 84.17 ± 0.22 | 81.77 ± 0.32 | 67.24 ± 1.63 | 82.20 ± 2.18 | 87.60 ± 0.29 | 87.54 ± 0.26 | 71.42 ± 4.43 | 73.68 ± 4.06 |
| | Reddit | 62.32 ± 0.54 | 61.58 ± 0.72 | 63.40 ± 0.36 | 64.84 ± 0.84 | 63.65 ± 0.41 | 60.81 ± 0.26 | 64.49 ± 0.25 | 64.98 ± 0.20 | 65.35 ± 0.60 | 66.74 ± 0.13 |
| | MOOC | 63.22 ± 1.55 | 62.92 ± 1.24 | 76.72 ± 0.30 | 77.07 ± 3.40 | 74.69 ± 0.68 | 74.28 ± 0.53 | 73.99 ± 0.97 | 81.41 ± 0.31 | 80.82 ± 0.30 | 81.64 ± 0.67 |
| | LastFM | 70.39 ± 4.31 | 71.45 ± 1.75 | 76.28 ± 0.25 | 69.46 ± 4.65 | 71.33 ± 0.47 | 65.78 ± 0.65 | 76.42 ± 0.22 | 77.01 ± 0.43 | 76.35 ± 0.52 | 79.22 ± 0.33 |
| | Enron | 65.86 ± 3.71 | 62.08 ± 2.27 | 61.40 ± 1.30 | 62.90 ± 1.16 | 60.72 ± 0.36 | 67.11 ± 0.62 | 72.37 ± 1.38 | 72.85 ± 0.81 | 67.07 ± 0.62 | 75.12 ± 1.43 |
| | Social Evo. | 88.51 ± 0.87 | 88.72 ± 1.10 | 93.97 ± 0.54 | 90.65 ± 1.62 | 79.83 ± 0.39 | 94.10 ± 0.32 | 94.01 ± 0.47 | 96.91 ± 0.12 | 96.82 ± 0.17 | 95.45 ± 0.30 |
| | UCI | 63.16 ± 2.27 | 52.47 ± 2.09 | 70.49 ± 0.93 | 70.73 ± 0.79 | 64.54 ± 0.47 | 76.65 ± 0.99 | 81.64 ± 0.49 | 82.06 ± 0.58 | 72.13 ± 1.86 | 73.65 ± 3.70 |
| | Can. Parl. | 52.58 ± 0.86 | 52.24 ± 0.28 | 56.46 ± 0.50 | 54.18 ± 0.73 | 57.06 ± 0.08 | 55.46 ± 0.69 | 55.76 ± 0.65 | N/A | 87.22 ± 0.82 | 91.59 ± 1.10 |
| | US Legis. | 52.94 ± 2.11 | 62.10 ± 1.41 | 51.83 ± 3.95 | 61.18 ± 1.10 | 55.56 ± 1.71 | 53.87 ± 1.41 | 52.03 ± 1.02 | N/A | 56.31 ± 3.46 | 55.90 ± 0.74 |
| | UN Trade | 55.43 ± 1.20 | 55.42 ± 0.87 | 55.58 ± 0.68 | 52.80 ± 3.24 | 54.97 ± 0.38 | 55.66 ± 0.98 | 54.88 ± 1.01 | N/A | 52.56 ± 1.70 | 56.80 ± 0.27 |
| | UN Vote | 61.17 ± 1.33 | 60.29 ± 1.79 | 53.08 ± 3.10 | 63.71 ± 2.97 | 48.01 ± 0.82 | 54.13 ± 2.16 | 48.10 ± 0.40 | N/A | 52.61 ± 1.25 | 58.46 ± 0.91 |
| | Contact | 90.43 ± 2.33 | 89.22 ± 0.65 | 94.14 ± 0.45 | 88.12 ± 1.50 | 74.19 ± 0.81 | 90.43 ± 0.17 | 89.91 ± 0.36 | N/A | 93.55 ± 0.52 | 93.66 ± 0.56 |
| | Avg. Rank | 6.29 | 6.58 | 4.83 | 5.08 | 6.75 | 4.88 | 4.58 | N/A | 3.83 | 2.17 |

Table 17: AUC-ROC for inductive dynamic link prediction with random, historical, and inductive negative sampling strategies.

| | Datasets | JODIE | DyRep | TGAT | TGN | CAWN | TCL | GraphMixer | FreeDyG | DyGFormer | DyG-Mamba |
|---|---|---|---|---|---|---|---|---|---|---|---|
| rnd | Wikipedia | 94.33 ± 0.27 | 91.49 ± 0.45 | 95.90 ± 0.09 | 97.72 ± 0.03 | 98.03 ± 0.04 | 95.57 ± 0.20 | 96.30 ± 0.04 | 99.01 ± 0.02 | 98.48 ± 0.03 | 98.55 ± 0.01 |
| | Reddit | 96.52 ± 0.13 | 96.05 ± 0.12 | 96.98 ± 0.04 | 97.39 ± 0.07 | 98.42 ± 0.02 | 93.80 ± 0.07 | 94.97 ± 0.02 | 98.84 ± 0.01 | 98.71 ± 0.01 | 98.91 ± 0.01 |
| | MOOC | 83.16 ± 1.30 | 84.03 ± 0.49 | 86.84 ± 0.17 | 91.24 ± 0.99 | 81.86 ± 0.25 | 81.43 ± 0.19 | 82.77 ± 0.24 | 87.01 ± 0.74 | 87.62 ± 0.51 | 89.98 ± 0.46 |
| | LastFM | 81.13 ± 3.39 | 82.24 ± 1.51 | 76.99 ± 0.29 | 82.61 ± 3.15 | 87.82 ± 0.12 | 70.84 ± 0.85 | 80.37 ± 0.18 | 94.32 ± 0.03 | 94.08 ± 0.08 | 94.82 ± 0.01 |
| | Enron | 81.96 ± 1.34 | 76.34 ± 4.20 | 64.63 ± 1.74 | 78.83 ± 1.11 | 87.02 ± 0.50 | 72.33 ± 0.99 | 76.51 ± 0.71 | 89.51 ± 0.20 | 90.69 ± 0.26 | 90.60 ± 0.11 |
| | Social Evo. | 93.70 ± 0.29 | 91.18 ± 0.49 | 93.41 ± 0.19 | 93.43 ± 0.59 | 84.73 ± 0.27 | 93.71 ± 0.18 | 94.09 ± 0.07 | 96.41 ± 0.07 | 95.29 ± 0.03 | 95.41 ± 0.01 |
| | UCI | 78.80 ± 0.94 | 58.08 ± 1.81 | 77.64 ± 0.38 | 86.68 ± 2.29 | 90.40 ± 0.11 | 84.49 ± 1.82 | 89.30 ± 0.57 | 93.01 ± 0.08 | 92.63 ± 0.13 | 92.05 ± 0.23 |
| | Can. Parl. | 53.81 ± 1.14 | 55.27 ± 0.49 | 56.51 ± 0.75 | 55.86 ± 0.75 | 58.83 ± 1.13 | 55.83 ± 1.07 | 58.32 ± 1.08 | N/A | 89.33 ± 0.48 | 97.11 ± 0.09 |
| | US Legis. | 58.12 ± 2.35 | 61.07 ± 0.56 | 48.27 ± 3.50 | 62.38 ± 0.48 | 51.49 ± 1.13 | 50.43 ± 1.48 | 47.20 ± 0.89 | N/A | 53.21 ± 3.04 | 52.73 ± 1.24 |
| | UN Trade | 62.28 ± 0.50 | 58.82 ± 0.98 | 62.72 ± 0.12 | 59.99 ± 3.50 | 67.05 ± 0.21 | 63.76 ± 0.07 | 63.48 ± 0.37 | N/A | 67.25 ± 1.05 | 69.37 ± 0.06 |
| | UN Vote | 58.13 ± 1.43 | 55.13 ± 3.46 | 51.83 ± 1.35 | 61.23 ± 2.71 | 48.34 ± 0.76 | 50.51 ± 1.05 | 50.04 ± 0.86 | N/A | 56.73 ± 0.69 | 60.03 ± 0.02 |
| | Contact | 95.37 ± 0.92 | 91.89 ± 0.38 | 96.53 ± 0.10 | 94.84 ± 0.75 | 89.07 ± 0.34 | 93.05 ± 0.09 | 92.83 ± 0.05 | N/A | 98.30 ± 0.02 | 98.33 ± 0.01 |
| | Avg. Rank | 5.67 | 6.83 | 6.25 | 4.25 | 5.17 | 6.92 | 6.08 | N/A | 2.25 | 1.67 |
| hist | Wikipedia | 61.86 ± 0.53 | 57.54 ± 1.09 | 78.38 ± 0.20 | 75.75 ± 0.29 | 62.04 ± 0.65 | 79.79 ± 0.96 | 82.87 ± 0.21 | 82.08 ± 0.32 | 68.33 ± 2.82 | 69.73 ± 0.48 |
| | Reddit | 61.69 ± 0.39 | 60.45 ± 0.37 | 64.43 ± 0.27 | 64.55 ± 0.50 | 64.94 ± 0.21 | 61.36 ± 0.26 | 64.27 ± 0.13 | 66.79 ± 0.31 | 64.81 ± 0.25 | 67.82 ± 0.30 |
| | MOOC | 64.48 ± 1.64 | 64.23 ± 1.29 | 74.08 ± 0.27 | 77.69 ± 3.55 | 71.68 ± 0.94 | 69.82 ± 0.32 | 72.53 ± 0.84 | 81.52 ± 0.37 | 80.77 ± 0.63 | 82.74 ± 0.75 |
| | LastFM | 68.44 ± 3.26 | 68.79 ± 1.08 | 69.89 ± 0.28 | 66.99 ± 5.62 | 67.69 ± 0.24 | 55.88 ± 1.85 | 70.07 ± 0.20 | 72.63 ± 0.16 | 70.73 ± 0.37 | 72.52 ± 0.31 |
| | Enron | 65.32 ± 3.57 | 61.50 ± 2.50 | 57.84 ± 2.18 | 62.68 ± 1.09 | 62.25 ± 0.40 | 64.06 ± 1.02 | 68.20 ± 1.62 | 70.09 ± 0.65 | 65.78 ± 0.42 | 75.35 ± 1.06 |
| | Social Evo. | 88.53 ± 0.55 | 87.93 ± 1.05 | 91.87 ± 0.72 | 92.10 ± 1.22 | 83.54 ± 0.24 | 93.28 ± 0.60 | 93.62 ± 0.35 | 96.94 ± 0.17 | 96.91 ± 0.09 | 96.03 ± 0.24 |
| | UCI | 60.24 ± 1.94 | 51.25 ± 2.37 | 62.32 ± 1.18 | 62.69 ± 0.90 | 56.39 ± 0.47 | 66.54 ± 0.47 | 75.98 ± 0.84 | 76.01 ± 0.75 | 65.55 ± 1.01 | 66.93 ± 2.56 |
| | Can. Parl. | 51.62 ± 1.00 | 52.38 ± 0.46 | 58.30 ± 0.61 | 55.64 ± 0.54 | 60.11 ± 0.48 | 57.30 ± 1.03 | 56.68 ± 1.20 | N/A | 88.68 ± 0.74 | 91.54 ± 0.39 |
| | US Legis. | 58.12 ± 2.94 | 67.94 ± 0.98 | 49.99 ± 4.88 | 64.87 ± 1.65 | 54.41 ± 1.31 | 52.12 ± 2.13 | 49.28 ± 0.86 | N/A | 56.57 ± 3.22 | 56.15 ± 0.15 |
| | UN Trade | 58.73 ± 1.19 | 57.90 ± 1.33 | 59.74 ± 0.59 | 55.61 ± 3.54 | 60.95 ± 0.80 | 61.12 ± 0.97 | 59.88 ± 1.17 | N/A | 58.46 ± 1.65 | 62.81 ± 0.21 |
| | UN Vote | 65.16 ± 1.28 | 63.98 ± 2.12 | 51.73 ± 4.12 | 68.59 ± 3.11 | 48.01 ± 1.77 | 54.66 ± 2.11 | 45.49 ± 0.42 | N/A | 53.85 ± 2.02 | 62.69 ± 1.23 |
| | Contact | 90.80 ± 1.18 | 88.88 ± 0.68 | 93.76 ± 0.41 | 88.84 ± 1.39 | 74.79 ± 0.37 | 90.37 ± 0.16 | 90.04 ± 0.29 | N/A | 94.14 ± 0.26 | 94.18 ± 0.36 |
| | Avg. Rank | 5.92 | 7.00 | 5.33 | 5.17 | 6.25 | 5.08 | 4.58 | N/A | 3.50 | 2.17 |
| ind | Wikipedia | 61.87 ± 0.53 | 57.54 ± 1.09 | 78.38 ± 0.20 | 75.76 ± 0.29 | 62.02 ± 0.65 | 79.79 ± 0.96 | 82.88 ± 0.21 | 83.17 ± 0.31 | 68.33 ± 2.82 | 69.73 ± 0.48 |
| | Reddit | 61.69 ± 0.39 | 60.44 ± 0.37 | 64.39 ± 0.27 | 64.55 ± 0.50 | 64.91 ± 0.21 | 61.36 ± 0.26 | 64.27 ± 0.13 | 66.78 ± 0.36 | 64.51 ± 0.19 | 67.82 ± 0.30 |
| | MOOC | 64.48 ± 1.64 | 64.22 ± 1.29 | 74.07 ± 0.27 | 77.68 ± 3.55 | 71.69 ± 0.94 | 69.83 ± 0.32 | 72.52 ± 0.84 | 75.81 ± 0.69 | 80.77 ± 0.63 | 82.74 ± 0.75 |
| | LastFM | 68.44 ± 3.26 | 68.79 ± 1.08 | 69.89 ± 0.28 | 66.99 ± 5.61 | 67.68 ± 0.24 | 55.88 ± 1.85 | 70.07 ± 0.20 | 71.42 ± 0.33 | 70.73 ± 0.37 | 72.52 ± 0.31 |
| | Enron | 65.32 ± 3.57 | 61.50 ± 2.50 | 57.83 ± 2.18 | 62.68 ± 1.09 | 62.27 ± 0.40 | 64.05 ± 1.02 | 68.19 ± 1.63 | 68.79 ± 0.91 | 65.79 ± 0.42 | 75.35 ± 1.06 |
| | Social Evo. | 88.53 ± 0.55 | 87.93 ± 1.05 | 91.88 ± 0.72 | 92.10 ± 1.22 | 83.54 ± 0.24 | 93.28 ± 0.60 | 93.62 ± 0.35 | 96.79 ± 0.17 | 96.91 ± 0.09 | 96.03 ± 0.24 |
| | UCI | 60.27 ± 1.94 | 51.26 ± 2.40 | 62.29 ± 1.17 | 62.66 ± 0.91 | 56.39 ± 0.11 | 70.42 ± 1.93 | 75.97 ± 0.85 | 73.41 ± 0.88 | 65.58 ± 1.00 | 66.93 ± 2.56 |
| | Can. Parl. | 51.61 ± 0.98 | 52.35 ± 0.52 | 58.15 ± 0.62 | 55.43 ± 0.42 | 60.01 ± 0.47 | 56.88 ± 0.93 | 56.63 ± 1.09 | N/A | 88.51 ± 0.73 | 91.54 ± 0.39 |
| | US Legis. | 58.12 ± 2.94 | 67.94 ± 0.98 | 49.99 ± 4.88 | 64.87 ± 1.65 | 54.41 ± 1.31 | 52.12 ± 2.13 | 49.28 ± 0.86 | N/A | 56.57 ± 3.22 | 56.15 ± 0.15 |
| | UN Trade | 58.71 ± 1.20 | 57.87 ± 1.36 | 59.98 ± 0.59 | 55.62 ± 3.59 | 60.88 ± 0.79 | 61.01 ± 0.93 | 59.71 ± 1.17 | N/A | 57.28 ± 3.06 | 62.81 ± 0.21 |
| | UN Vote | 65.29 ± 1.30 | 64.10 ± 2.10 | 51.78 ± 4.14 | 68.58 ± 3.08 | 48.04 ± 1.76 | 54.65 ± 2.20 | 45.57 ± 0.41 | N/A | 53.87 ± 2.01 | 62.69 ± 1.23 |
| | Contact | 90.80 ± 1.18 | 88.87 ± 0.67 | 93.76 ± 0.40 | 88.85 ± 1.39 | 74.79 ± 0.38 | 90.37 ± 0.16 | 90.04 ± 0.29 | N/A | 94.14 ± 0.26 | 94.18 ± 0.36 |
| | Avg. Rank | 5.92 | 6.92 | 5.25 | 5.17 | 6.33 | 5.08 | 4.67 | N/A | 3.42 | 2.25 |

# E Proofs for Theorems

## E.1 Proofs of Theorem 1

*Proof.* Consider the following linear time-invariant (LTI) system on the interval $[t_{k-1}, t_k]$, where $t_k - t_{k-1} = \Delta t_{k,i}$ for the $i$-th coordinate:

$$\frac{d\boldsymbol{h}(s)}{ds} = \boldsymbol{A}_k\,\boldsymbol{h}(s) + \boldsymbol{B}_k\,\boldsymbol{m}_k, \quad \boldsymbol{h}(t_{k-1}) = \boldsymbol{h}_{k-1},$$

where $\boldsymbol{A}_k = \mathrm{diag}(\lambda_1, \dots, \lambda_n)$ and $\boldsymbol{B}_k = \begin{bmatrix}\boldsymbol{B}_{k,1}, \dots, \boldsymbol{B}_{k,n}\end{bmatrix}^\top$ (also arranged diagonally for each coordinate). Since $\boldsymbol{A}_k$ is diagonal with $\mathrm{Re}(\lambda_i) < 0$, the fundamental matrix solution for this system (i.e., the matrix exponential) is also diagonal and can be written as

$$e^{\boldsymbol{A}_k \Delta t_k} = \mathrm{diag}\Big(e^{\lambda_1 \Delta t_{k,1}}, \dots, e^{\lambda_n \Delta t_{k,n}}\Big).$$

**Step 1: Expressing $\overline{\boldsymbol{A}}_k$.**
By definition of the matrix exponential for a diagonal matrix:

$$\overline{\boldsymbol{A}}_k = e^{\boldsymbol{A}_k \Delta t_k} = \mathrm{diag}\Big(e^{\lambda_1 \Delta t_{k,1}}, \dots, e^{\lambda_n \Delta t_{k,n}}\Big).$$

**Step 2: Expressing $\overline{\boldsymbol{B}}_k$.**
We compute the convolution term associated with the inhomogeneous part $\boldsymbol{B}_k\,\boldsymbol{m}_k$. For a diagonal system, the solution for each coordinate $i$ is:

$$\boldsymbol{h}_{k,i} = e^{\lambda_i \Delta t_{k,i}}\,\boldsymbol{h}_{k-1,i} + \int_0^{\Delta t_{k,i}} e^{\lambda_i (\Delta t_{k,i} - \tau)}\,\boldsymbol{B}_{k,i}\,\boldsymbol{m}_{k,i}\,d\tau.$$

Since $\boldsymbol{m}_{k,i}$ is constant w.r.t. $\tau$ and $\boldsymbol{B}_{k,i}$ is also constant in this interval, we can pull them out of the integral:

$$\boldsymbol{h}_{k,i} = e^{\lambda_i \Delta t_{k,i}}\,\boldsymbol{h}_{k-1,i} + \boldsymbol{B}_{k,i}\,\boldsymbol{m}_{k,i} \int_0^{\Delta t_{k,i}} e^{\lambda_i (\Delta t_{k,i} - \tau)}\,d\tau.$$

Evaluating the integral:

$$\int_0^{\Delta t_{k,i}} e^{\lambda_i (\Delta t_{k,i} - \tau)}\,d\tau = \int_0^{\Delta t_{k,i}} e^{\lambda_i (\Delta t_{k,i} - u)}\,du = \left[ -\frac{1}{\lambda_i}\,e^{\lambda_i (\Delta t_{k,i} - u)} \right]_0^{\Delta t_{k,i}} = \frac{1}{\lambda_i}\left(e^{\lambda_i \Delta t_{k,i}} - 1\right).$$

Hence,

$$\boldsymbol{h}_{k,i} = e^{\lambda_i \Delta t_{k,i}}\,\boldsymbol{h}_{k-1,i} + \frac{1}{\lambda_i}\left(e^{\lambda_i \Delta t_{k,i}} - 1\right)\boldsymbol{B}_{k,i}\,\boldsymbol{m}_{k,i}.$$

This leads to

$$\overline{\boldsymbol{B}}_k = \mathrm{diag}\Big(\lambda_1^{-1}\big(e^{\lambda_1 \Delta t_{k,1}} - 1\big)\boldsymbol{B}_{k,1}, \dots, \lambda_n^{-1}\big(e^{\lambda_n \Delta t_{k,n}} - 1\big)\boldsymbol{B}_{k,n}\Big),$$

since each diagonal entry of $\overline{\boldsymbol{B}}_k$ matches the integral factor $\lambda_i^{-1}(e^{\lambda_i \Delta t_{k,i}} - 1)\boldsymbol{B}_{k,i}$.

**Step 3: Final Form of $\boldsymbol{h}_{k,i}$.**
Combining the above results yields the stated coordinate-wise update for $\boldsymbol{h}_{k,i}$:

$$\boldsymbol{h}_{k,i} = e^{\lambda_i \Delta t_{k,i}}\,\boldsymbol{h}_{k-1,i} + \lambda_i^{-1}\left(e^{\lambda_i \Delta t_{k,i}} - 1\right)\boldsymbol{B}_{k,i}\,\boldsymbol{m}_{k,i}.$$

This confirms the forms of both $\overline{\boldsymbol{A}}_k$ and $\overline{\boldsymbol{B}}_k$ as well as the final expression for each coordinate $\boldsymbol{h}_{k,i}$.

$\square$

## E.2 Proofs of Theorem 2

*Proof.* Since DyG-Mamba has three core parameters, *i.e.*, $\Delta$, $\boldsymbol{B}$ and $\boldsymbol{C}$ that control its effectiveness. We define Theorem 2 to explain the main effects of the parameters $\boldsymbol{B}$ and $\boldsymbol{C}$.

We first copy Eq.(8) from the paper, SSM-based node representation learning process, as follows

$$\boldsymbol{h}_k = \bar{\boldsymbol{A}}_k \boldsymbol{h}_{k-1} + \bar{\boldsymbol{B}}_k \boldsymbol{u}_k, \quad \widehat{\boldsymbol{z}}_k^\tau = \bar{\boldsymbol{C}}_k \boldsymbol{h}_k, \tag{17}$$

$h_k$ in Eq.(17) can be further decomposed as follows:

$$h_k = \prod_{i=0}^{k-2} \bar{A}_{k-i}\bar{B}_1 u_1 + \cdots + \prod_{i=0}^{k-j} \bar{A}_{k-i}\bar{B}_{j-1} u_{j-1} + \cdots + \bar{B}_k u_k, \tag{18}$$

Then, considering $A$ is one fixed parameter and $\bar{A}_k = \exp(\Delta t_k A)$, $\hat{z}_k^\tau$ in Eq.(8) can be formulated as:

$$\begin{aligned}
\hat{z}_k^\tau &= \bar{C}_k \prod_{i=0}^{k-2} \bar{A}_{k-i}\bar{B}_1 u_1 + \cdots + \bar{C}_k \prod_{i=0}^{k-j} \bar{A}_{k-i}\bar{B}_{j-1} u_{j-1} + \cdots + \bar{C}_k \bar{B}_k u_k, \\
&= e^{(\sum_{i=0}^{k-2} \Delta t_{k-i} A)} \bar{C}_k \bar{B}_1 u_1 + \cdots + e^{(\sum_{i=0}^{k-j-1} \Delta t_{k-i} A)} \bar{C}_k \bar{B}_j u_j + \cdots + \bar{C}_k \bar{B}_k u_k,
\end{aligned} \tag{19}$$

where $u_k$ is the k-th input of $M_u^\tau$, i.e., $u_k = M_u^\tau[k, :]$, and parameters $B = \text{Linear}_B(M_u^\tau)$ and $C = \text{Linear}_C(M_u^\tau)$. Thus, $\bar{C}_k$ can be considered as one query of $k$-th input $u_k$ and $\bar{B}_j$ can be considered as the key of $j$-th input $u_j$. Then, $\bar{C}_k \bar{B}_j$ can measure the similarity between $u_k$ and $u_j$, similar to the self-attention mechanism. According to the above-mentioned proof, we can conclude that parameters $B$ and $C$ can measure the similarity between current input to the previous ones and selectively copy the previous input. $\qquad\square$

### E.3 Proofs of Theorem 3

*Proof.* **Step 1: Parameter perturbation bounds**. Under spectral normalization constraints ($\|W_B\|_2 \le 1$, $\|W_C\|_2 \le 1$), the parameter perturbations induced by input noise $\Delta M_u^\tau$ satisfy:

$$\|\Delta B\| \le \|W_B\|_2 \|\Delta M_u^\tau\| \le \|\Delta M_u^\tau\|, \quad \|\Delta C\| \le \|W_C\|_2 \|\Delta M_u^\tau\| \le \|\Delta M_u^\tau\|. \tag{20}$$

This follows directly from the Lipschitz continuity enforced by spectral normalization, where the spectral norm $\|W\|_2$ precisely defines the maximum amplification factor of the linear transformation.

**Step 2: Perturbation propagation analysis**. From the state update decomposition in Theorem 4.2, the output perturbation $\Delta \widehat{m}_k^\tau$ can be expressed as:

$$\|\Delta \widehat{m}_k^\tau\| \le \sum_{j=1}^{k} \left( \|\Delta \overline{C}_k\| \|\overline{B}_j\| + \|\overline{C}_k\| \|\Delta \overline{B}_j\| \right) \|m_j\| \prod_{i=0}^{k-j-1} \|e^{\Delta t_{k-i} A_{k-i}}\|. \tag{21}$$

Using $\|\overline{B}_j\|, \|\overline{C}_k\| \le 1$ (spectral normalization) and $\|\Delta \overline{B}_j\|, \|\Delta \overline{C}_k\| \le \|\Delta M_u^\tau\|$ (Step 1), we derive:

$$\|\Delta \widehat{m}_k^\tau\| \le \|\Delta M_u^\tau\| \sum_{j=1}^{k} e^{\gamma(t_k - t_j)}. \tag{22}$$

Here, the exponential term $\prod_{i=0}^{k-j-1} \|e^{\Delta t_{k-i} A_{k-i}}\| \le e^{\gamma(t_k - t_j)}$ arises from the eigenvalue constraint $\gamma = \max_i \text{Re}(\lambda_i(A)) < 0$.

**Step 3: Stability via exponential decay**. For $\gamma < 0$, the summation over exponentially decaying terms can be approximated as:

$$\sum_{j=1}^{k} e^{\gamma(t_k - t_j)} \approx \int_0^T e^{\gamma(T-t)} dt = \frac{1}{|\gamma|} \left(1 - e^{\gamma T}\right), \tag{23}$$

where $T$ is the total sequence duration. This yields the final perturbation bound:

$$\|\Delta \widehat{m}_k^\tau\| \le \kappa \|\Delta M_u^\tau\|, \quad \text{where } \kappa = \frac{1}{|\gamma|}(1 - e^{\gamma T}). \tag{24}$$

$\qquad\square$

## Impact Statement

This paper aims to advance the field of Machine Learning by introducing a Mamba-based framework for continuous-time dynamic graph modeling. We examine its potential impacts from two primary perspectives as follows. *(i) dynamic graph modeling*. With the rapid expansion of social and economic networks, dynamic graph modeling has emerged as a prominent research topic in the machine learning community. In contrast to existing methods, our approach introduces a novel Mamba-based framework, which incorporates irregular timespans as control signals for continuous SSMs. This design enhances the model's ability to effectively and efficiently capture long-term temporal dependency on dynamic graphs. *(ii) Time Information Effects*. Despite significant advances in dynamic graph modeling, there is still a lack of theoretical foundations regarding the influence of temporal information on the evolution of dynamic graphs. Our work highlights the significant potential of the timespan-based memory forgetting mechanism to deepen the theoretical understanding of time in this domain. Overall, we do not foresee any direct negative societal implications resulting from this research. Although more advanced modeling capabilities can be applied across a range of domains, we believe that this methodology itself does not introduce any ethical or societal concerns beyond those typically associated with improvements in machine learning.

