# OpenReview forum: "DyG-Mamba: Continuous State Space Modeling on Dynamic Graphs"
_NeurIPS.cc/2025/Conference — NeurIPS 2025 poster_

### Official Review · Reviewer_Cc3i · 2025-06-24

**Clarity:** 3
**Significance:** 3
**Originality:** 3
**Rating:** 5
**Confidence:** 3

**Summary:**

This paper introduces DyG-Mamba, a novel continuous state space model (SSM) tailored for dynamic graph modeling. By drawing inspiration from Ebbinghaus forgetting and review cycles, the model introduces a timespan-informed mechanism to dynamically forget or reinforce historical information. DyG-Mamba redefines the core SSM parameters—Δt, A, B, and C—to handle irregular timespans and filter noisy data, thereby aiming to improve the effectiveness, robustness, and efficiency of dynamic graph modeling. The authors evaluate DyG-Mamba on 12 benchmark datasets across tasks like link prediction and node classification, reporting SOTA performance in accuracy, speed, memory, and robustness.

**Questions:**

1. What does the performance look like on smaller graphs with short sequences? Does DyG-Mamba still justify its complexity?
2. The paper reports linear complexity but test on sequences ≤2,048.  Can DyG-Mamba handle billion-edge graphs (e.g., social networks)?  If not, what are the bottlenecks?

**Ethical Concerns:**

["NO or VERY MINOR ethics concerns only"]

**Final Justification:**

I wish to express my sincere appreciation for the authors' thorough and substantive rebuttal.

The supplementary experiments on small-scale datasets (Table 1) convincingly establish DyG-Mamba’s adaptability beyond long sequences, while scalability validation on the tgbl-coin-v2 benchmark (22.8M edges) substantiates linear efficiency claims. Furthermore, the theoretical refinements to metaphorical descriptions and expanded comparisons against contemporaneous works (e.g., Ding et al.) significantly strengthen the paper’s technical framing.

Conditional upon seamless integration of these new results, expanded discussions, and comparative analyses into the final manuscript, I endorse this work for acceptance.

**Paper Formatting Concerns:**

No major formatting issues were identified in the submission.

**Quality:**

3

**Strengths And Weaknesses:**

#### Strengths:

1. The paper proposes a timespan-informed continuous SSM for dynamic graph modeling, integrating inspiration from psychological memory theories, which is an intuitive and creative angle.
2. DyG-Mamba redefines core SSM parameters (∆t, A, B, C) to enable time-aware forgetting and input-conditioned reviewing, aiming to improve robustness and generalization — a technically meaningful contribution.
3. The model demonstrates strong empirical performance across 12 datasets, with results exceeding or matching state-of-the-art baselines under various negative sampling schemes.

#### Weaknesses:
1. The metaphorical explanations (e.g., Ebbinghaus forgetting/review curves) are somewhat overemphasized and can feel stretched when mapped to a technical context.
2. The comparison with recent SSM-based dynamic graph models (e.g., DG-Mamba) is missing, even though such models target similar goals.
3. While the model is described as lightweight, the added architectural complexity (learnable ∆t function, spectral constraints, dynamic B and C) may increase training difficulty without clear discussion of stability or convergence behavior.

---

> ### Author Rebuttal · Authors · 2025-07-31
>
> **Thank you very much for your valuable and constructive comments. We sincerely appreciate your efforts to help improve the quality of our work. We have carefully responded to your concerns and hope that our revisions will help alleviate them.**
>
> *Q1. What does the performance look like on smaller graphs with short sequences...*
>
> A1. Thank you for your thoughtful question. We appreciate your concern regarding the applicability of DyG-Mamba to smaller graphs with shorter temporal sequences. To address this, we conducted additional experiments and provided detailed empirical comparisons, which will be included in our paper to enhance clarity. Specifically, we evaluate DyG-Mamba on three relatively small-scale datasets—UCI, Can. Parl., and US Legis.—under short input sequence lengths of 32 and 64. We compare DyG-Mamba with two strong baselines GraphMixer and DyGFormer as follows:
>
> Table 1. Performance (AP) on smaller graphs with short input sequences.
>
> |Model|UCI|LastFM|Enron|
> |-|-|-|-|
> |GraphMixer-32|93.25±0.18|75.60±0.25|82.23±0.03|
> |GraphMixer-64|93.12±0.22|75.61±0.28|82.21±0.04|
> |DyGFormer-32|95.79±0.17|89.12±0.15|83.51±0.06|
> |DyGFormer-64|95.76±0.15|90.86±0.14|83.74±0.05|
> |DyG-Mamba-32|96.79±0.08|89.22±0.17|86.28±0.06|
> |DyG-Mamba-64|96.83±0.05|91.12±0.13|88.43±0.05|
> |
>
> As shown in the table, **DyG-Mamba consistently outperforms both baselines, even on small graphs with short input sequences.** This suggests that our model effectively adapts to different input sequence lengths, not just long-term modeling. We believe this is because short sequences can still contain irregular timestamps and noise, which often degrade the performance of baselines. In contrast, DyG-Mamba leverages timespan-aware memory updates and selective noise filtering to address these challenges. We hope this explanation helps alleviate your concerns.
>
> *Q2. The paper...Can DyG-Mamba handle billion-edge graphs...*
>
> A2. Thank you for raising this important point. Although we noted the dataset scale as a limitation in our paper, we fully agree with your suggestion that demonstrating performance on larger graphs is essential to support our scalability claims. To further verify the scalability and efficiency of DyG-Mamba on million-edge temporal graphs, we conducted additional experiments on tgbl-coin-v2 [1], which contains 638K nodes and 22.8M temporal edges. We strictly followed the standardized training pipeline and hyperparameter setup as proposed in [2] to ensure fair comparison. The results are shown in Table below:
>
> |Method|Performance|Epoch Running Time (s)| GPU Usage (M)|
> |-|-|-|-|
> |DyRep| 45.20±4.60|178,719|48,336|
> |TGN|58.60±3.70|138,408|48,116|
> |GraphMixer|75.31±0.21|43,160|12,204|
> |DyGFormer|75.17±0.38|163,151|41,348|
> |DyG-Mamba|76.14±0.30|45,124|18,094|
> |
>
> As the table illustrates, **DyG-Mamba not only achieves superior predictive performance but also outperforms the best-performing baseline in training efficiency, further validating its scalability on real-world large-scale dynamic graphs.** We sincerely apologize for not including experiments on billion-edge graphs. This is mainly due to the current lack of publicly available benchmark datasets of that scale in the continuous-time dynamic graph domain. We acknowledge this as a limitation and will explicitly add this point to the revised version of our paper.
>
> [1] Shenyang, Huang et al. Temporal graph benchmark for machine learning on temporal graphs. NeurIPS 2023.\
> [2] Le Yu. An Empirical Evaluation of Temporal Graph Benchmark. ArXiv, 2023.
>
> *Weakness-1: The metaphorical explanations are somewhat overemphasized and can feel stretched when mapped to a technical context.*
>
> A3. Thank you for your thoughtful suggestion. We understand your concern regarding the potentially overemphasized metaphorical explanations. Our intention was to use these metaphors as intuitive motivation to help readers better understand the rationale behind our temporal modeling design. Following your suggestion, we will revise the original descriptions to be more concise and technically focused, ensuring that the metaphorical elements serve only as high-level intuition without overshadowing our core technical contributions. Specifically, **we will streamline and modify Lines 152-157 regarding the Ebbinghaus forgetting curve as follows**:
> - The Ebbinghaus forgetting curve can be formulated as R=exp⁡(−t/S) [1,2], where R denotes memory retention, t is the time interval, and S is a decay constant. Inspired by this formulation, we reinterpret R as a timespan-dependent decay coefficient, allowing the model to apply temporal decay to historical states proportionally to the elapsed timespan.
>
> In addition, we will revise the metaphorical explanations in Introduction to make them more concise and technically focused. We sincerely hope that the revisions we have committed to will help resolve your concerns.
>
> [1] Wozniak, P. et al. Two components of long-term memory. Acta neurobiologiae experimentalis, 2010.\
> [2] Wixted, T. The psychology and neuroscience of forgetting. Annu. Rev. Psychol., 2004.
>
> *Weakness-2: The comparison with recent SSM-based dynamic graph models (e.g., DG-Mamba) is missing, even though such models target similar goals.*
>
> A4. We sincerely thank the reviewer for pointing out the missing comparison with DG-Mamba [1]. We would like to clarify that while DG-Mamba is indeed a recent SSM-based model for dynamic graphs, it is specifically designed for discrete-time dynamic graphs where the input is segmented into predefined snapshots with fixed or uniform time intervals. In contrast, our work targets continuous-time dynamic graphs, where events occur at irregular time intervals, and thus demands a fundamentally different temporal modeling approach. Although we cited DG-Mamba in Lines 85–86 of the original manuscript, we acknowledge that our explanation may have been too brief and could have caused confusion. To address this, **we plan to revise Lines 85–86 in the paper to clearly state the scope and limitations of DG-Mamba and why it is not directly comparable in our experiments as follows**：
> - DG-Mamba [1] and GraphSSM [2] extend Mamba to discrete-time dynamic graphs by modeling snapshot sequences with fixed time intervals. However, these methods are not applicable to continuous-time dynamic graphs with irregular timestamps, and thus are not directly comparable to our setting.
>
> To further alleviate your concerns, we additionally include a recent concurrent work, Ding et al. [3], which adapts Mamba for continuous-time dynamic graph modeling, as a new baseline for comparison. Ding et al. propose a heuristic framework that stacks node-level and time-level Mamba blocks to capture edge-specific temporal patterns for link prediction. We reproduce this baseline following the settings and best-tuned hyperparameters reported in their paper, and report the performance (AP) in the table below. And our method consistently outperforms Ding et al.[3] across multiple datasets, demonstrating the effectiveness of our proposed Mamba adaptation specifically tailored for dynamic graphs.
>
> |Benchmarks|LastFM|Enron|MOOC|Reddit|Wikipedia|UCI|Social Evo.|
> |-|-|-|-|-|-|-|-|
> |Ding et al. [3]|93.26±0.02|92.56±0.04|89.12±0.15|99.13±0.02|99.00±0.01|95.35±0.09|94.12±0.01|
> |DyG-Mamba|94.22±0.04|93.22±0.03|90.17±0.19|99.25±0.00|99.06±0.01|96.79±0.08|94.75±0.01|
> |
>
> We sincerely hope that the additional comparisons and corresponding revisions will help alleviate your concerns regarding the selection of baselines.
>
> [1] Haonan Yuan et al., DG-Mamba: Robust and Efficient Dynamic Graph Structure Learning with Selective State Space Models. AAAI, 2024.\
> [2] Jintang Li et al., State space models on temporal graphs: A first-principles study. NeurIPS, 2024.\
> [3] Zifeng Ding et al., Efficiently Modeling Long-Term Temporal Dependency on Continuous-Time Dynamic Graphs with State Space Models. TMLR, 2025.
>
> *Weakness-3: While the model is described as lightweight, the added architectural complexity (learnable ∆t function, spectral constraints, dynamic B and C) may increase training difficulty without clear discussion of stability or convergence behavior.*
>
> A5. We appreciate your insightful comment regarding training difficulty. To address this, we will include clarifying discussion in our revised version. These task-specific designs are lightweight and stable in practice, as detailed below.
> - Learnable ∆t Function：We implement ∆t as an element-wise exponential decay function, realized by a scalar-wise MLP. This function is differentiable, non-recurrent, and incurs negligible computational overhead.
> - Spectral Norm Constraint: This serves as a regularization to prevent the B and C matrices from exploding, without introducing any additional parameters. As proven in Theorem 3, this design ensures bounded output under input noise, thereby enhancing both robustness and convergence stability.
> - Input-Dependent B and C: We re-define B and C as input-dependent parameters using shallow, non-recurrent linear layers. This structure introduces minimal additional parameters and computational cost while enabling the model to selectively attend to relevant inputs. To further support our training efficiency, we show smooth loss curves and consistent convergence behavior on Can.Parl. dataset as follows:
> |Model|Epoch=20|Epoch=40|Epoch=60|Epoch=80|Epoch=100|
> |-|-|-|-|-|-|
> | Vanilla Mamba|0.2229|0.2026|0.1964|0.1922|0.1914|
> |DyG-Mamba|0.1691|0.1478|0.1480|0.1482|0.1480|
> |
>
> As shown in the table, our method converges within 100 epochs, demonstrating fast and stable training behavior. We observe similar convergence patterns across all other datasets as well.  We hope these additional details clearly address your concerns regarding Weakness-3, particularly the model's training stability.
>
> **We sincerely hope our response has helped address your concerns. If you feel that all issues have been resolved, we would truly appreciate your support in the final evaluation.**

---

> > ### Comment · Reviewer_Cc3i · 2025-08-04
> >
> > I wish to express my sincere appreciation for the authors' thorough and substantive rebuttal.
> >
> > The supplementary experiments on small-scale datasets (Table 1) convincingly establish DyG-Mamba’s adaptability beyond long sequences, while scalability validation on the tgbl-coin-v2 benchmark (22.8M edges) substantiates linear efficiency claims.         Furthermore, the theoretical refinements to metaphorical descriptions and expanded comparisons against contemporaneous works (e.g., Ding et al.) significantly strengthen the paper’s technical framing.
> >
> > Conditional upon seamless integration of these new results, expanded discussions, and comparative analyses into the final manuscript, I endorse this work for acceptance.

---

> ### Author Response · Authors · 2025-08-04
> **Acknowledgement to Reviewer Cc3i**
>
> We would like to express our heartfelt appreciation for the time and thoughtful attention you dedicated to reviewing our work, as well as for your constructive and insightful comments.
>
> We are committed to integrating the additional discussions and experimental results into the final version of our manuscript to further enhance its clarity, depth, and overall quality.
>
> We are especially grateful for your recognition of our contributions to the dynamic graph community and your recommendation for acceptance. Your endorsement serves as a strong encouragement for us to continue advancing our research in this direction.
>
> **Kind Reminder: Thank you again for endorsing our work for acceptance. We noticed that the final rating has not yet been submitted, so please don’t forget to submit it when it’s convenient for you.** 😊

---

### Official Review · Reviewer_NsaC · 2025-06-28

**Clarity:** 3
**Significance:** 2
**Originality:** 2
**Rating:** 4
**Confidence:** 4

**Summary:**

DyG-Mamba reframes continuous-time dynamic graph modeling as long-sequence modeling with a selective SSM. It treats the irregular timespan between events as a control signal, redefining the SSM’s step size and transition matrix so forgetting follows an Ebbinghaus-style curve. To counteract over-forgetting and noise, it makes the projection parameters input-dependent and applies spectral-norm constraints, letting the model selectively “review” important history.  This design yields linear-time and memory complexity, enabling full-length sequences without pooling.  Across 12 benchmarks in link prediction and node classification, DyG-Mamba attains state-of-the-art accuracy, is up to 8.9× faster, and uses 77 % less GPU memory than Transformer baselines, while remaining robust to up to 60 % noisy edges.

**Questions:**

None

**Ethical Concerns:**

["NO or VERY MINOR ethics concerns only"]

**Final Justification:**

Most of my concerns have been addressed. I still believe that DyG-Mamba is more akin to a mamba extension of DyGFormer, and the scope of this work should be improved from this perspective. Given the new results and extra effort the authors have made, I'd like to increase the rating for the paper.

**Limitations:**

Yes.

**Quality:**

3

**Strengths And Weaknesses:**

**Strengths**:

1. The authors reformulate dynamic-graph modeling as a continuous SSM and give formal guarantees on stability, forgetting behaviour (Theorem 4.1), and noise attenuation (Theorem 4.3).

2. The experiments are comprehensive, demonstrating convincing performance compared with various baseline methods.

3. Given the ability of the SSM backbone that scales linearly with sequence length, the proposed method opens the door for much more efficient implementation for many downstream tasks.

**Weakness**:

1. The choice of the specific learnable function for Δt in Equation (11) seems to be important to the method. The paper does not provide a detailed justification or ablation for this specific mathematical form versus other possible monotonically increasing functions. Was this form empirically chosen, and how sensitive is the model to this choice?

2. The paper lacks a detailed discussion on how the theoretical bound in Theorem 4.3 is related to the empirical evaluation in Section 5.4.

3. Experiments remain on modest-scale graphs; it is unclear whether the linear scan kernel remains GPU-efficient on million-edge streams or with heterogeneous edge types.

4. My overall feeling of the paper is to use Mamba architecture on DyGFormer and address the same problem that DyGFormer worked on. Most of the advantages the method exhibits are inherent in the Mamba architecture. Although it is good to see that the authors made improvements on SSM to tailor it for the DGL task, it is still hard to find highlights in the paper.

---

> ### Author Rebuttal · Authors · 2025-07-31
>
> **We sincerely thank you for your constructive and thoughtful feedback. We truly appreciate your recognition of our theoretical contributions and comprehensive experiments. We will address each of your concerns in detail and revise the manuscript accordingly.**
>
> *Q1. The choice of the specific learnable function for Δt in Equation (11) seems to be important to the method. The paper does not provide a detailed justification or ablation for this specific mathematical form versus other possible monotonically increasing functions. Was this form empirically chosen, and how sensitive is the model to this choice?*
>
> A1. We thank you for raising this insightful question. The design of the learnable function for Δt is inspired by the Ebbinghaus forgetting curve [1,2]. Specifically, our formulation Eq.(11) mirrors the mathematical form of the Ebbinghaus forgetting curve as proposed in [3,4]: R=exp⁡(−t/S), where R denotes memory retention, t is the time interval, and S is a decay constant. In our design, we reinterpret the retention function R as part of the timespan-dependent scaling factor to ensure that larger timespans induce stronger decay, aligning the model behavior with human memory dynamics. Following your suggestion, we provide several alternative monotonic functions and conduct ablation studies to validate our design. These results will be included in our paper:
> - (1) Linear Variant: Δt = $w ⊙ (\frac{t_{k+1}-t_{k}}{\tau-t_{1}})$.
> - (2) Logarithmic Variant: Δt = $w_1$ ⊙ log$(1+ w_{2} ⊙ \frac{t_{k+1}-t_{k}}{\tau-t_{1}})$.
> - (3) Sigmoid Variant: Δt = $w_{1}$ ⊙ Sigmoid$(w_{2} ⊙ \frac{t_{k+1}-t_{k}}{\tau-t_{1}})$.
> - (4) Exponential Variant: Δt = $w_{1}$ ⊙ exp$(w_{2} ⊙ \frac{t_{k+1}-t_{k}}{\tau-t_{1}})$.
> - (5) w/o $(\tau-t_{1})$: remove normalization item in Eq.(11).
> - (6) w/o timespan: replaces timespans with input samples u(t) as control signals, i.e., Δt =SiLU(Linear(u(t))).
>
> |Variants|Can. Parl.|Enron|USLegis.|
> |-|-|-|-|
> | Linear|94.64±0.18|91.13±0.06|70.28±1.84|
> | Log|97.18±0.12|92.35±0.05|72.46±2.34|
> | Sigmoid|95.84±0.13|92.26±0.04|72.15±2.36|
> | Exp| 94.68±0.11|90.84±0.06|71.23±1.48|
> | w/o ($\tau-t_{1}$)|97.32±0.20|92.46±0.16|72.45±0.84|
> | w/o timespan|96.90±0.18|92.14±0.12|72.26±0.76|
> | DyG-Mamba|98.37±0.07|93.22±0.03|74.11±2.23|
> |
>
> These results demonstrate that our DyG-Mamba achieves superior performance. Compared to the Linear and Log variants, our decay function better models diminishing memory influence with a smoother and bounded curve. The Sigmoid variant suffers from early saturation, limiting its ability to differentiate mid- and long-range intervals. The Exp variant over-amplifies long timespans, making it sensitive to noise. Moreover, removing normalization or timespan leads to performance degradation, confirming the necessity of our full design. We hope this response addresses your concerns. We appreciate your insightful suggestions and will incorporate this ablation study and its analysis into the revised paper to help readers better understand the motivation and effectiveness behind our Δt design.
>
> [1] Über das gedächtnis, Duncker & Humbolt, 1885.\
> [2] Replication and analysis of Ebbinghaus’ forgetting curve. PLoS ONE, 2015.\
> [3] Two components of long-term memory. Acta neurobiologiae experimentalis, 2010.\
> [4] The psychology and neuroscience of forgetting. Annu. Rev. Psychol., 2004.
>
> *Q2. The paper lacks a detailed discussion on how the theoretical bound in Theorem 4.3 is related to the empirical evaluation in Section 5.4.*
>
> A2. Thank you for your insightful question. We will explicitly strengthen the discussion linking Theorem 4.3 to the empirical robustness results in Section 5.4 to help readers better understand its practical implications. Specifically, Theorem 4.3 provides a theoretical robustness guarantee by bounding the output perturbation under input noise. Section 5.4 empirically supports this claim by injecting random noisy edges into the input historical sequences and evaluating model performance. DyG-Mamba shows only minor performance degradation compared to baselines, demonstrating its strong resistance to noise. To further strengthen this connection, we include two additional experiments:
> - (1) Lipschitz Bound Tightness: We examine whether the theoretical bound in Theorem 4.3 accurately captures the empirical output deviation under input perturbations. Using the Wikipedia dataset, we report the input/output perturbation, the bound, and whether it holds. These results confirm that the empirical behavior of DyG-Mamba adheres to the theoretical bounds.
>
> | Noise Ratio | Input Perturbation | Output Perturbation | Theoretical Bound | Within Bound? | Performance |
> |-|-|-|-|-|-|
> |10%|0.13|0.14|0.16|Yes|0.987|
> |20%|0.27|0.30|0.33|Yes|0.985 |
> |30%|0.39|0.44|0.48|Yes|0.982 |
> |40%|0.48|0.54|0.60|Yes|0.973 |
> |50%|0.57|0.65|0.71|Yes|0.968 |
> |60%|0.66|0.74|0.82|Yes|0.958 |
> |
>
> - Ablation Study: We further demonstrate that Theorem 4.3 plays a critical role in enhancing the model's robustness. Here, we use Can. Parl. dataset as a case study.
>
> | Baselines/Noise Ratio (%)|0|10| 20| 30| 40| 50|
> |-|-|-|-|-|-|-|
> | w/o Theorem 4.3| 96.65±0.13|96.25±0.11|95.75±0.10| 95.25±0.08|94.83±0.08|93.21±0.11|
> | DyG-Mamba|98.37±0.07|98.32±0.06|98.05±0.05|97.95±0.05|97.90±0.07|97.74±0.06|
> |
>
> In summary, Theorem 4.3 explains DyG-Mamba’s strong noise robustness. Our new experiments confirm the bound’s tightness and show that removing it leads to clear performance drops. We will add these results in our paper to better highlight the practical value of Theorem 4.3.
>
>
> *Q3. Experiments...it is unclear whether the linear scan kernel remains GPU-efficient on million-edge streams or with heterogeneous edge types.*
>
> A3. Thank you for your valuable suggestion. To further verify the scalability and efficiency of DyG-Mamba on million-edge dynamic graphs with heterogeneous structures, we conduct additional experiments on the large-scale tgbl-coin-v2 [1], which consists of 638K nodes and 22.8M temporal edges. tgbl-coin-v2 includes heterogeneous edge types, with transactions from five stablecoins and one wrapped token categorized into two types to reflect different modes of fund transfer [2]. This setting enables us to evaluate DyG-Mamba’s capacity to handle both large-scale and heterogeneous temporal interactions. For a fair comparison, we follow the training details as listed in [3] and adopt the best-performing hyperparameters for all baselines. The performance (MRR) of DyG-Mamba and strong baselines on tgbl-coin-v2 can be summarized below:
>
> |Method|Performance|Epoch Running Time (s)|GPU Usage (M)|
> |-|-|-|-|
> |DyRep|45.20±4.60|178,719|48,336|
> |TGN|58.60±3.70|138,408|48,116|
> |GraphMixer|75.31±0.21|43,160|12,204|
> |DyGFormer|75.17±0.38|163,151|41,348|
> |DyG-Mamba|76.14±0.30|45,124|18,094|
> |
>
> These results show that DyG-Mamba achieves the best performance on the heterogeneous tgbl-coin-v2 dataset, highlighting its strong modeling capability under multi-type temporal interactions. In addition, compared to DyGFormer—another strong baseline designed for long-range dependency modeling—DyG-Mamba exhibits significantly lower runtime and memory consumption while delivering superior performance. We will include these results and the corresponding discussion in the revised version to further illustrate DyG-Mamba’s scalability and practical efficiency.
>
> [1] Temporal graph benchmark for machine learning on temporal graphs. NeurIPS, 2023.\
> [2] Labeled graph datasets for UTXO and account-based blockchains. NeurIPS, 2022.\
> [3] An Empirical Evaluation of Temporal Graph Benchmark. arXiv, 2023.
>
> *Q4. My overall feeling...it is still hard to find highlights in the paper.*
>
> A4. Thank you for raising this thoughtful question.  We fully understand your concerns and would like to explain that DyG-Mamba is not merely an architecture replacement of DyGFormer. Instead, DyG-Mamba introduces (1) a new inductive bias by the Ebbinghaus forgetting curve to guide memory updates; (2) a novel timespan function for effectively modeling irregular temporal patterns; (3) a linearly scalable architecture with theoretically grounded noise attenuation and empirically validated robustness.
> We fully agree that Mamba’s core strengths—such as its linear-time complexity—form a strong foundation for DyG-Mamba. However, the main contribution lies in adapting and extending the SSM to address the unique challenges of dynamic graph learning. The "highlights" of our paper could be summarized as follows:
> - **Ebbinghaus-Inspired Timespan Function for Irregular Temporal Modeling**: We extend the continuous SSM to dynamic graphs by incorporating irregular timespans as key control signals. Inspired by the Ebbinghaus forgetting curve, we design a learnable timespan function that redefines Δt and guides memory decay through initialization of the transition matrix A. This enables effective modeling of irregular temporal patterns that vanilla SSMs like Mamba do not address.
> - **Robust Memory Control via Input-Dependent Parameterization**: Inspired by Ebbinghaus review cycles, we make the core SSM parameters B and C input-dependent and regulate their capacity using spectral norm constraints. This enables DyG-Mamba to selectively revisit relevant history while suppressing noisy or irrelevant signals, enhancing robustness in noisy environments.
> - **Addressing DGL-Specific Challenges**: RNN-based methods suffer from gradient issues, while Transformer-based ones face quadratic complexity. DyG-Mamba overcomes both with a linear-time architecture enhanced by timespan-aware mechanisms. Through selective memory updates and input-adaptive parameterization, DyG-Mamba mitigates the impact of noisy or irrelevant historical signals, improving stability in real-world dynamic graphs.
>
> **We sincerely hope that our explanation helps alleviate your concerns and provides a clearer understanding of our work. Could you kindly raise your score and champion our paper if you think all concerns are addressed ?**

---

> ### Author Response · Authors · 2025-08-05
> **Acknowledgement to Reviewer NsaC**
>
> Dear Reviewer NsaC
>
> **Thank you so much for your encouraging feedback and for your support toward the acceptance of our paper.** 😊
>
> Yes, DyG-Mamba is a principled redesign of dynamic graph modeling using state space models (SSMs). But it is never easy to adapt SSMs for continuous-time dynamic graph learning.
> - We take inspiration from the Ebbinghaus forgetting curve to explicitly control the SSM parameter updates based on the timespan information.
> - Strong theoretical guarantees are provided to explain why this design can be successful for continuous and long-term dynamic graph modeling.
> - Comprehensive evaluations demonstrate the effectiveness, inductiveness and robustness of our method.
>
> Thus, humbly, we would like to take this opportunity to say: DyG-Mamba represents a significant leap forward, offering a new paradigm to the graph/dynamic graph community.
>
> Best regards,
>
> Authors of DyG-Mamba

---

### Official Review · Reviewer_ps9a · 2025-07-02

**Clarity:** 3
**Significance:** 3
**Originality:** 3
**Rating:** 5
**Confidence:** 3

**Summary:**

This paper introduces a Mamba-based approach to continuous-time dynamic graphs, with (with mainly) experiments on dynamic link prediction. It is the first state-space model applied to the continuous-graph domain, and the idea is very promising.

**Questions:**

The performance increase appears marginal; why is this the case? More discussion should be added to explore this phenomenon.

Can't also the DyG-Former learn long-term dependencies by using a larger patch size?

How does the model behave when applied to high-frequency temporal networks?

Some work in the continuous-time graph literature, for example, the Piecewise-Velocity Model for Learning Continuous-Time Dynamic Node Representations [1], is missing from the related-work section. Could it be connected to the proposed SSM (especially as an first-order SSM )?

[1] Piecewise-Velocity Model for Learning Continuous-Time Dynamic Node Representations.

**Ethical Concerns:**

["NO or VERY MINOR ethics concerns only"]

**Final Justification:**

The paper makes a novel contribution by extending state-space models to the challenging setting of continuous-time dynamic networks, a problem of broad interest to the graph learning community.

The theoretical development is rigorous, and the authors present a comprehensive experimental setup that demonstrates the model’s robustness even under high-noise conditions.

The rebuttal clarified several key points; the additional details provided there should be integrated into the final manuscript.

I also encourage the authors to include a controlled study on synthetic networks. Such an experiment would allow them to vary interaction frequency and noise systematically, highlighting the scenarios in which the framework excels.

**Limitations:**

Yes.

**Paper Formatting Concerns:**

Format follows the NeurIPS 2025 Paper Instructions.

**Quality:**

3

**Strengths And Weaknesses:**

**Strengths**:

The paper provides a linear-time SSM model, and the idea of applying such methods to continuous-time graphs is both novel and very promising. The design choice of including two SSM blocks, considering both node-level and edge-level information, is also interesting.

There is solid theoretical depth to the paper, and the connection with Ebbinghaus’ review cycle makes the approach principled and well-motivated.

Results on link-prediction experiments are quite good, and the node-classification challenge is also discussed.

Importantly, the paper introduces an extensive experimental set-up, which demonstrates that the DyG-Mamba structure significantly improves computational and memory efficiency compared with the Transformer.

The paper studies multiple negative-sampling strategies, adding value and completeness.

The case study offers valuable insights regarding the comparison with the DyG-Former.

**Weaknesses**

The central idea is that, for dynamic graphs, especially those in the arduous continuous-time category, a model should learn by examining the long history of the sequence, which can now be extended because the model’s memory requirements do not need extra aggregation. Looking at Figure 2, panel (A) shows at best a marginal, or not even substantial, performance increase, while the message in panel (B) is much clearer. It is unclear what happens with the other networks; one could argue that, for some datasets, relying on a long history is unnecessary. As I see it, the model is not able to distinguish between these two cases.

The introduction of Δt, connected to the Ebbinghaus curve, is an elegant way to model the decay of historical impact with all its attractive theoretical properties. This choice, however, has the drawback that the decay does not rely on the actual content of the sequence: if an important signal is temporally distant, it could be disregarded. This raises the question, or necessity, of additional gating functions.

Although this limitation is acknowledged in the paper, the networks included are limited in both size and domain, making some of the main claims about long-history modelling not easily generalisable from the current experimental set-up.

---

> ### Author Rebuttal · Authors · 2025-07-31
>
> **Thank you very much for your valuable and constructive comments. We sincerely appreciate your efforts to help improve the quality of our work.**
>
> *Q1. The performance increase appears marginal...More discussion should be added to explore this phenomenon.*
>
> A1. Thank you for your insightful question. We agree that the performance gain of DyG-Mamba appears relatively modest on certain datasets. This is because those datasets are dominated by high-frequency and short-range interaction patterns, which limit the benefit of long-term modeling. For example, in Wikipedia, user-page edit interactions often occur in rapid bursts within seconds. In such settings, recent interactions already provide sufficient predictive signals, making it difficult for long-term modeling to deliver substantial improvement. As a result, short-term models already capture most useful signals, leaving limited room for DyG-Mamba’s long-term memory to contribute further.
>
> In contrast, DyG-Mamba shows more substantial improvements on datasets such as Enron, UCI and Can. Parl., which exhibit  longer-range and irregular interaction patterns. For example, Enron captures corporate email communications over several months, where interactions are less frequent and often separated by long time gaps. In these settings, modeling long-term and irregular dynamics is crucial. DyG-Mamba is specifically designed to address these challenges, allowing it to capture informative long-term signals while suppressing noisy or outdated ones.
>
> We hope this explanation clarifies the performance variations across datasets and provides better insight into the strengths of DyG-Mamba. We will include this analysis in our paper to help readers interpret the results more effectively.
>
> *Q2. Can't DyG-Former...by using a larger patch size?*
>
> A2. Thank you for your thoughtful question. We provide additional experiments to investigate how patch size affects DyG-Former’s performance. Specifically, we fixed the input sequence length to 512 and varied the patch size on LastFM. The results are shown below:
>
> |Patch Size|8|16|32|64|128|
> |-|-|-|-|-|-|
> | DyGFormer (AP)|92.14±0.11|93.00±0.12|92.03±0.14|91.26±0.16|90.33±0.13|
> |
>
> As shown in the table, performance consistently degrades as patch size increases. We attribute this to several factors:
> - **Temporal Coarsening**: Larger patches aggregate multiple events into a single unit, which blurs fine-grained temporal patterns essential for modeling continuous-time dynamics.
> - **Uniform Attention Assumption**: DyGFormer assigns equal importance to all patches and lacks mechanisms to adaptively focus on temporally irregular or informative regions. This limits its ability to capture the most relevant historical signals, especially in real-world scenarios with irregular timespans and noisy interactions.
> - **Sensitivity to Noise**: Larger patches are more likely to include irrelevant or noisy events, which further undermines performance.
>
> We appreciate your suggestion and will include this discussion in the appendix to help readers better understand the limitations of the baseline DyGFormer.
>
> *Q3. How does the model behave when applied to high-frequency temporal networks?*
>
> A3. Thank you for your insightful question. In our experiments, Wikipedia, Reddit and Contact are high-frequency temporal networks, where interactions occur at second-level intervals (Table 5). Although DyG-Mamba consistently outperforms baselines on these datasets, the performance gains over baselines tend to be modest. This is because most predictive signals are embedded in recent interactions, and while long-range dependencies still contribute, their marginal benefit is reduced due to the temporal density and redundancy of recent events. To further investigate this, we conducted an ablation study by varying the historical window size while keeping all other settings fixed. The results (AP) are summarized below:
>
> |Dataset|32|64|128|256|512|1024|2048|
> |-|-|-|-|-|-|-|-|
> |Wikipedia|98.98±0.02|99.06±0.01|99.08±0.01|99.10±0.02|99.10±0.01|99.12±0.02|99.16±0.03|
> |Reddit|99.11±0.01|99.25±0.00|99.25±0.01|99.28±0.01|99.28±0.01|99.30±0.02|99.33±0.02|
> |
>
> We find that increasing the sequence length yields only marginal improvements, indicating that while DyG-Mamba effectively leverages long-term information, the nature of high-frequency graphs limits the additional benefits from extended temporal modeling. We appreciate the reviewer’s question and will include this empirical analysis in our paper to help readers better understand DyG-Mamba's behavior in high-frequency dynamic graphs.
>
> *Q4. Some work...is missing from the related-work.*
>
> A4. We thank the reviewer for pointing out this relevant work and we will include it in the final version: PIVEM [1] learns dynamic node embeddings by approximating temporal evolution by piecewise linear interpolation, based on a latent distance model with piecewise constant and node-specific velocities. It can be viewed as a special case of a first-order SSM, where the hidden state corresponds to node velocity and evolves linearly over time.
>
> *Weakness-1. The central idea...to distinguish between these two cases.*
>
> A5. Thank you for your comment. The effectiveness of long-term modeling indeed varies by dataset. In high-frequency graphs like Wikipedia or Reddit, where interactions occur within seconds, useful information is concentrated in the short-term context. As shown in Table 1, increasing the input sequence length provides only marginal improvement because distant history adds little additional signal. In contrast, datasets such as Enron and Can. Parl. contain sparse, irregular interactions with longer temporal gaps. DyG-Mamba benefits more in these settings by leveraging its timespan-aware decay and memory update mechanisms to capture long-term dependencies while filtering out temporal noise.
>
> Table 1 Performance (AP) of DyG-Mamba with varying input length.
> |Dataset|32|64|128|256|512|1024|
> |-|-|-|-|-|-|-|
> |Wikipedia|98.98±0.02|99.06±0.01|99.08±0.01|99.10±0.02|99.10±0.01|99.12±0.02|
> |Reddit|99.11±0.01|99.25±0.00|99.25±0.01|99.28±0.01|99.28±0.01|99.30±0.02|
> |Enron| 86.28±0.06|88.43±0.05|89.95±0.05|91.19±0.04|93.22±0.03|93.58±0.06|
> |Can. Parl.| 76.22±0.57|77.12±0.28|77.96±0.35|78.35±0.27|94.83±0.07|98.19±0.06|
> |
>
> We also compared DyG-Mamba with GraphMixer, a strong baseline that focuses on short-term input sequence settings. In Table 2, DyG-Mamba achieves notable performance gains, which shows that DyG-Mamba’s time-aware decay mechanism and noise-resistant memory updates provide value even on high-frequency graphs.
>
> Table 2  Performance comparison on high-frequency temporal graphs.
> |Dataset|Wikipedia|Reddit|Contact|LastFM|
> |-|-|-|-|-|
> |GraphMixer|97.25±0.03|97.31±0.01|91.92±0.03|75.61±0.24|
> |DyG-Mamba|99.06±0.01|99.25±0.00|98.37±0.01|94.22±0.04|
> |
>
> DyG-Mamba dynamically adjusts attention based on temporal distance and relevance, which allows it to extract meaningful history only when needed. We will revise the paper to clarify the dataset-dependent impact of long-term modeling and highlight DyG-Mamba’s adaptability to different temporal patterns.
>
> *Weakness-2. The introduction of Δt...raises the question or necessity of additional gating functions.*
>
> A6. Thank you for highlighting this important point. We fully agree that a purely time-based decay may overlook semantically important yet temporal distant events. To address this, DyG-Mamba incorporates a form of content-aware gating via its input-dependent parameterization of the core matrices B and C (Theorem 4.2). These matrices function similarly to Query and Key in Transformer, dynamically controlling how much a historical input contributes based on its semantic alignment with the current context. Specifically, while the exponential decay term exp(f(Δt)) reduces the weight of distant events, the actual impact of a past event is jointly determined by its content via the interaction term C and B in Eq.(12).  When a temporally distant input is semantically important, it can still have a strong impact. We appreciate this suggestion, and to avoid misunderstanding, we will revise the manuscript to explicitly highlight how our architecture achieves this implicit gating behavior.
>
> *Weakness-3. Although this limitation...from the current experimental set-up.*
>
> A7. Thank you for your valuable suggestion. We acknowledge the importance of evaluating model generalizability across larger and more diverse networks. To this end, we conduct additional experiments on the large-scale tgbl-coin-v2 [1], which consists of 638K nodes and 22.8M temporal edges. tgbl-coin-v2 includes heterogeneous edge types, with transactions from five stablecoins and one wrapped token categorized into two types to reflect different modes of fund transfer [2]. The performance of DyG-Mamba and strong baselines on tgbl-coin-v2 is summarized below:
> |Method|Performance|Epoch Running Time (s)| GPU Usage (M)|
> |-|-|-|-|
> |DyRep| 45.20±4.60|178,719|48,336|
> |TGN|58.60±3.70|138,408|48,116|
> |GraphMixer|75.31±0.21|43,160|12,204|
> |DyGFormer|75.17±0.38|163,151|41,348|
> |DyG-Mamba|76.14±0.30|45,124|18,094|
> |
>
> These results show that DyG-Mamba achieves the best performance (MRR) on large-scale datasets, further supporting its ability to handle long-term temporal dependencies. We believe this extended evaluation helps address concerns about the scale and domain diversity. We will incorporate this result and its discussion into the revised version to strengthen the empirical validation of our method.
>
> **We sincerely hope our detailed responses and additional experiments have addressed your concerns and clarified the strengths of our work. If you find our revisions satisfactory, we would greatly appreciate it if you could consider raising your score and supporting our paper.**
>
>
> [1] Temporal graph benchmark for machine learning on temporal graphs. NeurIPS, 2023. \
> [2] Labeled graph datasets for UTXO and account-based blockchains. NeurIPS, 2022.

---

> > ### Comment · Reviewer_ps9a · 2025-08-01
> > **Response to authors**
> >
> > I would like to thank the authors for their detailed response, it is much appreciated.
> >
> > A lot of my concerns have been sufficiently addressed.
> >
> > Under the condition that these results and explanations will be integrated to the main paper, I would like to support the acceptance of the paper.
> >
> > I have one final recommendation for the authors: to incorporate an artificial-dataset experimental setup that lets you control the frequency and range of interactions. This would make it easier to demonstrate the proposed framework’s capabilities and to highlight the scenarios in which it delivers the greatest advantage.

---

> ### Author Response · Authors · 2025-08-01
> **Acknowledgement to Reviewer ps9a**
>
> Thank you very much for your kind and thoughtful feedback. We sincerely promise to carefully incorporate all the suggested changes into our paper.
>
> We also greatly appreciate your valuable suggestion regarding the use of artificial datasets. In future work, we plan to design a suitable synthetic setup to better demonstrate how interaction frequency influences the performance of our proposed method.
>
> Finally, we sincerely appreciate your recognition of our contributions to the dynamic graph community and your support for the acceptance of our paper.

---

### Official Review · Reviewer_kbHS · 2025-07-03

**Clarity:** 2
**Significance:** 2
**Originality:** 2
**Rating:** 3
**Confidence:** 2

**Summary:**

This paper proposes DyG-Mamba, a new approach for analyzing dynamic graphs. The method adapts state space models by using time gaps between events as control signals. It draws inspiration from Ebbinghaus' forgetting curve, which shows that people forget things quickly at first, then more slowly over time. The model works similarly by adjusting how much it forgets based on time intervals. It also learns to keep important past information while filtering out noise. The main benefit is speed and memory efficiency. DyG-Mamba achieves top results with linear complexity instead of quadratic, making it much faster than transformer-based methods. It also stays robust even when there's a lot of noise in the data. This efficiency makes it possible to handle much larger and longer graph sequences, which could be useful for social networks and recommendation systems.

**Questions:**

1. The paper only looks at adding edges, but real graphs are more complex - nodes appear and disappear, features change over time. How would you extend DyG-Mamba to handle these cases? Are there basic limitations with using SSMs for node changes?
2. Your biggest dataset has 2.4M edges, but real networks like Facebook or Twitter are much larger. Do you have any evidence this actually works on bigger graphs? Even tests on fake large graphs would help. The linear complexity sounds good but the hidden constants might be huge.
3. Why use spectral norm constraints of exactly 1.0? What happens if you change or remove them? The ablation study doesn't really test this. Some basic sensitivity tests would show if this choice actually matters.
4. The case study is too simple. Can you show clearer examples of what the model chooses to remember versus forget? How does it handle when the data patterns change over time?
5. Why no comparison with Graph Neural ODEs that also work with continuous time? Also, copying results from other papers instead of running the methods yourself makes it hard to trust the comparisons.

**Main concerns:** The core idea seems reasonable but needs more proof. I'd want to see: (a) tests on realistic graph sizes, (b) better analysis of design choices, (c) clearer understanding of what the model actually does. Right now the scope feels too narrow to be convincing.

**Ethical Concerns:**

["NO or VERY MINOR ethics concerns only"]

**Limitations:**

Yes.

**Paper Formatting Concerns:**

No.

**Quality:**

2

**Strengths And Weaknesses:**

Strengths:

1. This paper introduces the first use of state space models for continuous-time dynamic graphs, which is a solid contribution. The approach of using time gaps as control signals is clever and well-motivated by psychological research on memory. The three theorems provide theoretical backing for the design choices, and the linear complexity is a clear advantage over transformer methods.

2. The experiments cover 12 datasets with multiple evaluation settings, which is thorough. The results are generally strong across different tasks and show good consistency. The efficiency gains are substantial - much faster training and lower memory usage compared to existing methods.

3. The model handles noisy data well, maintaining performance even with 60% noise injection. This is important for real applications where data quality varies. The ablation studies help understand which components matter most.

4. The paper is well-written overall. The connection between memory theories and the technical approach is explained clearly, and the figures help illustrate the main ideas.

Weaknesses:

1. The biggest limitation is that experiments only cover edge addition on relatively small graphs. Real applications involve much larger networks with node changes, feature evolution, and different graph types. The largest dataset has only 2.4M edges, which is small by modern standards.

2. While the linear complexity sounds good, the actual tests only go up to sequence length 2048. It's unclear how the method performs on much longer sequences or whether memory requirements become problematic. The constant factors in the complexity could be large.

3. Some baselines are dated, and newer graph ODE methods are missing. Results for some methods are copied from other papers rather than re-implemented, which raises questions about fair comparison. The evaluation doesn't include recent 2024 methods.

4. The case study doesn't provide much insight into what the model actually learns. It's hard to understand when the method might fail or how it handles changes in data patterns over time. More analysis of the learned representations would be helpful.

5. The paper lacks discussion of training stability, hyperparameter sensitivity, and deployment considerations. The evaluation is limited to link prediction and node classification, missing other important temporal tasks like anomaly detection or community tracking.

---

> ### Author Rebuttal · Authors · 2025-07-31
>
> **Thank you very much for your efforts and time to review our paper and give us many helpful comments! We have carefully addressed your comments and hope our responses help clarify our contributions.**
>
> *Q1 & Weak-1. The paper only looks at adding edges, but real graphs are more complex - nodes appear and disappear, features change over time...Are there basic limitations with using SSMs for node changes?*
>
> A1. We appreciate your comment. We would like to clarify that DyG-Mamba does include these complex settings and we will add these descriptions in our paper.
> - Node Changes: In continuous-time link prediction, nodes dynamically appear and disappear, since the model only observes the one-hop neighborhood within a recent time window (typically of length 32 to 1024) at each prediction step. As a result, the set of observed nodes is time-dependent and varies across prediction steps.
> - Feature Changes: In link prediction datasets, nodes always have different features at different timestamps. DyG-Mamba is capable of handling such time-varying node features.
> - Different Graph Types: In our experiments, (1) Wikipedia, Reddit, MOOC, LastFM are heterogeneous bipartite graphs; (2) Can. Parl., US Legis., UN Trade and UN Vote are dynamic snapshots; (3) Enron, Social Evo., UCI are continuous-time dynamic graphs.
>
> DyG-Mamba not only supports the settings described above, but is also robust to noisy real-world conditions (Figure 5). We hope this helps clarify our experimental design and alleviates your concerns.
>
> *Q2 & Weak-2. Your biggest dataset has 2.4M edges...Do you have any evidence this actually works on bigger graphs? Even tests on fake large graphs would help. The linear complexity sounds good but the hidden constants might be huge.*
>
> A2. Thank you for your suggestion, we add one large-scale benchmark tgbl-coin-v2 [1], which contains 638 K nodes and 22.8 M edges.  We hope the results alleviate your concerns regarding DyG-Mamba’s scalability on large-scale graphs.
>
> |Method|Performance|Epoch Running Time (s)| GPU Usage (M)|
> |-|-|-|-|
> |DyRep|45.20±4.60|178,719|48,336|
> |TGN|58.60±3.70|138,408|48,116|
> |GraphMixer|75.31±0.21|43,160|12,204|
> |DyGFormer|75.17±0.38|163,151|41,348|
> |DyG-Mamba|76.14±0.30|45,124|18,094|
> |
>
> We provide a more detailed analysis of runtime and memory usage on the Can.Parl. dataset. We hope the table below helps alleviate your concerns regarding the hidden constant in DyG-Mamba's complexity.
>
> |Input Length|256|512|1024|2048|
> |-|-|-|-|-|
> |DyG-Former (AP)|74.62±0.37|84.94±0.08|91.69±0.06|97.35±0.06|
> |DyGFormer Running Time/epoch (s)|324.40|487.55|630.44|946.99|
> |DyGFormer Memory Usage (M)|1,986|5,176|21,548|40,692|
> |DyG-Mamba (AP)|78.35±0.27|94.83±0.07|98.19±0.06|98.37±0.07|
> |DyG-Mamba Running Time/epoch (s)|160.32|168.26|174.45|190.13|
> |DyG-Mamba Memory Usage (M)|1,282|2,234|5,230|11,728|
> |
>
> As shown in the table,
> - Running time increases gradually, confirming the low constant factor in our linear-time implementation.
> - Memory usage grows linearly as expected, and remains within acceptable bounds even for long sequences.
> - Model performance (AP) remains stable across different lengths, indicating efficiency without compromising predictive power.
>
> We believe these results provide strong empirical evidence that the hidden constant in DyG-Mamba’s complexity is small. We will add this analysis to our revised paper to clarify the practical benefits of our design.
>
> *Q3. Why use spectral norm constraints of exactly 1.0? What happens if you change or remove them? The ablation study doesn't really test this. Some basic sensitivity tests would show if this choice actually matters.*
>
> A3.We thank the reviewer for the insightful question. The spectral norm constraint of $\leq$ 1 is a necessary theoretical requirement to ensure the robustness guarantee established in Theorem 3, as supported by the theoretical proof in Appendix E.3. Here, we provide additional ablations and promise to add this table in our paper:
> |Spectral Norm Bound|Can.Parl.|Enron|USLegis.|
> |-|-|-|-|
> | w/o constraints|96.65±0.13|92.06±0.12|73.24±0.75|
> | $\leq$ 2|96.98±0.07| 92.68±0.11|73.59±0.67|
> | $\leq$ 1.5|97.61±0.10|92.93±0.13|73.86±1.42|
> | DyG-Mamba ($\leq$ 1)|98.37±0.07|93.22±0.03|74.11±2.23|
> |
>
> As shown in the table, relaxing the spectral norm constraint leads to consistently lower performance across all datasets. In particular, removing the constraint entirely ("w/o constraints") results in the most noticeable drop, highlighting its necessity. Even modest relaxations (e.g., ≤2 or ≤1.5) yield inferior results compared to our default setting (≤1), suggesting that tighter control of the parameter magnitude contributes meaningfully to model stability and predictive performance.
> We sincerely thank the reviewer for this suggestion and commit to including this ablation study and its analysis in the revised version of our paper to better justify the use of spectral norm constraints in DyG-Mamba.
>
> *Q4 & Weak-4. The case study is too simple. Can you show clearer examples of what the model chooses to remember versus forget? How does it handle when the data patterns change over time?*
>
> A4. We thank the reviewer for this suggestion. We will revise our case study with a longer input sequence and clearer visualization to illustrate what DyG-Mamba chooses to remember versus forget. Specifically, DyG-Mamba tends to retain neighbor nodes that (i) frequently reappear in the historical sequence or (ii) are semantically similar to the target node, as these provide more predictive signals. In contrast, neighbors that are rarely observed or semantically unrelated are gradually down-weighted. When data patterns change over time, DyG-Mamba adapts by adjusting the memory decay rate through the learnable, timespan-aware parameter Δt (Eq. 11). This enables the model to quickly suppress outdated interactions and prioritize recent, meaningful signals. For example, in a transaction network, if a user suddenly begins interacting with new counterparties, DyG-Mamba assigns lower weights to outdated neighbors and increasingly attends to newly frequent ones. This allows the model to dynamically respond to behavioral drift without manual resetting. Additionally, the input-dependent transition and projection matrices (B and C), regularized via spectral norm constraints (Eq. 13), enhance the model’s ability to filter out transient or noisy inputs.
> We will incorporate this enhanced case study and detailed explanation in the revised paper to better showcase DyG-Mamba’s adaptive memory behavior.
>
>
> *Q5 & Weak-3. Why no comparison with Graph Neural ODEs that also work with continuous time? Also, copying results from other papers instead of running the methods yourself makes it hard to trust the comparisons.*
>
> A5. We thank the reviewer for highlighting the importance of comparing with Graph Neural ODE-based models. We have now included one SOTA graph neural ODE baseline CTAN [2] and two additional baselines [3,4] in our experiments. The AP performance is reported as follows:
>
> |Datasets|FreeDyG|CTAN|Ding et al.|DyG-Mamba|
> |-|-|-|-|-|
> |Wiki|99.23±0.01|98.92±0.01|99.00±0.01|99.06±0.01|
> |Reddit|98.99±0.00| 99.03±0.02|99.13±0.02|99.25±0.00|
> |MOOC|88.39±0.14| 88.58±0.13|89.12±0.15|90.17±0.19|
> |LastFM|90.28±0.30| 91.32±0.05|93.26±0.02|94.22±0.04|
> |Enron|91.53±0.09|91.88±0.04|92.56±0.04|93.22±0.03|
> |Social Evo.|94.54±0.01|94.21±0.01|94.12±0.01|94.75±0.01|
> |UCI| 95.88±0.12|95.31±0.08|95.35±0.09|96.79±0.08|
> |
>
> These results show that DyG-Mamba consistently outperforms ODE-based methods across all datasets, highlighting its strong generalization in continuous-time dynamic graphs. Regarding the concern about copied results, we confirm that all baseline results were independently reproduced by using the open-source benchmarking toolkit Dynamic Graph Library [5]. We follow standardized training protocols, shared preprocessing, and consistent hyperparameters (from DyGLib load_configs.py) and fix random seeds to ensure fair comparisons.
>
> *Weak-5. The paper lacks discussion of training stability, hyperparameter sensitivity, and deployment considerations. The evaluation is limited...*
>
> A6. We thank the reviewer for pointing out this important aspect. We will add training stability results to provide a more comprehensive view of DyG-Mamba’s behavior during optimization. In particular, we include below the training loss across epochs on the Can.Parl. dataset:
>
> | Model|Epoch=20|Epoch=40|Epoch=60|Epoch=80| Epoch=100|
> |-|-|-|-|-|-|
> |Vanilla Mamba|0.2229|0.2026|0.1964|0.1922|0.1914|
> |DyG-Mamba|0.1691|0.1478|0.1480|0.1482|0.1480|
> |
>
> As shown in the table, DyG-Mamba exhibits faster convergence and more stable loss reduction, completing training within 100 epochs. We observe similar trends across all other datasets, indicating the model’s training stability. In addition, hyperparameter sensitivity and deployment considerations have been discussed in the appendix. We will explicitly refer to these sections in the main paper for better visibility:
> - Appendix D.4 (Figure 9 and Table 11) presents hyperparameter sensitivity analysis.
> - Appendix D.2 (Tables 6, 7 and Figure 7) covers model size, memory usage, and runtime for deployment.
> - Limitation Section: We acknowledge that exploring more domains and tasks is an important direction for future work.
>
>
> **Could you kindly raise your score and champion our paper if you think all concerns are addressed ? Thank you again for your valuable comments and review.**
>
> [1] Huang, Shenyang, et al. Temporal graph benchmark for machine learning on temporal graphs. NeurIPS 2023.\
> [2] Long Range Propagation on Continuous-Time Dynamic Graphs. ICML, 2024.\
> [3] FreeDyG: Frequency Enhanced Continuous-Time Dynamic Graph Model for Link Prediction. ICLR-2024.\
> [4] Efficiently Modeling Long-Term Temporal Dependency on Continuous-Time Dynamic Graphs with State Space Models. TMLR, 2025.\
> [5] Towards Better Dynamic Graph Learning: New Architecture and Unified Library. NeurIPS, 2023.

---

### Author Response · Authors · 2025-08-08
**Summary of Reviews**

Dear Area Chair,

**Thank you so much for your great efforts. As the rebuttal phase draws to a close, for your convenience, we would like to summarize the reviews and rebuttals as follows.**

3 out of 4 reviewers acknowledged that our rebuttal well addressed their concerns and endorsed our work for acceptance. Unfortunately, the remaining 1 reviewer remained silent throughout the entire rebuttal and discussion phase, despite our comprehensive responses to their comments.

DyG-Mamba is a principled redesign of dynamic graph modeling using state space models (SSMs), by taking inspiration from the Ebbinghaus forgetting curve to explicitly control the SSM parameter updates based on the timespan information.

Strong theoretical guarantees are provided to explain why this design can be successful for continuous-time and long-term dynamic graph modeling. Comprehensive evaluations demonstrate the effectiveness, efficiency, inductiveness and robustness of our method.

Accordingly, we consider DyG-Mamba worthy of publication at NeurIPS and believe it will make a valuable contribution to the graph learning community.

Best Regards, Authors

---

### Note · Authors · 2025-08-11

Dear SACs and Area Chairs,

**Thank you so much for your great efforts. For your convenience, we would like to summarize the reviews, rebuttals and contributions  as follows.**

3 out of 4 reviewers acknowledged that our rebuttal well addressed their concerns and endorsed our work for acceptance. Unfortunately, the remaining 1 reviewer remained silent throughout the entire rebuttal and discussion phase, despite our comprehensive responses to their comments.

In the rebuttal, to directly address reviewers’ concerns, we further strengthened our work by
- **Scalability to Large-scale Graphs**: Reviewers questioned the model's scalability. We addressed this by introducing a new large-scale benchmark (tgbl-coin-v2) with 22.8M edges, where DyG-Mamba achieved SOTA performance, demonstrating its scalability.
- **Comparison to Recent Baselines**: Reviewers noted the lack of comparison with recent Continuous-Time GNNs. We added a comparison with SOTA baselines like CTAN and FreeDyG, showing that DyG-Mamba outperforms them on all datasets.
- **Sufficiency of Core Design Choices**: Questions were raised about key design elements, such as the spectral norm constraint and the timespan function. We provided new ablation studies that theoretically and empirically validated these choices, showing significant performance drops when they were removed.


In summary, the **main contributions of DyG-Mamba** are as follows:

- DyG-Mamba is a principled redesign of dynamic graph modeling using state space models (SSMs), by taking inspiration from the Ebbinghaus forgetting curve to explicitly control the SSM parameter updates based on the timespan information.
- DyG-Mamba provides strong theoretical guarantees explaining why this design can be successful for continuous-time and long-term dynamic graph modeling.
- Comprehensive evaluations demonstrate the effectiveness, efficiency, inductiveness and robustness of our method.

Accordingly, we consider DyG-Mamba worthy of publication at NeurIPS and believe it will make a valuable contribution to the graph learning community.

Best Regards, Authors

---

### Decision · Program_Chairs · 2025-09-17

**Decision:**

Accept (poster)

**Comment:**

This paper introduces DyG-Mamba, the first state-space model applied to continuous-time dynamic graphs, which reframes dynamic graph modelling as long-sequence modelling with a selective state-space model (SSM). The approach treats irregular time intervals between events as control signals that redefine the SSM's step size and transition matrix to implement Ebbinghaus-style forgetting curves, while using input-dependent projection parameters and spectral-norm constraints to selectively "review" important historical information and counteract over-forgetting and noise. DyG-Mamba achieves linear-time and memory complexity, enabling processing of full-length sequences without pooling, and demonstrates state-of-the-art performance across 12 benchmark datasets on tasks like dynamic link prediction and node classification, excelling in accuracy, speed, memory efficiency, and robustness.

Reviewers felt that the paper makes a novel and theoretically rigorous contribution by extending state-space models to continuous-time dynamic graphs, with solid theoretical grounding that connects Ebbinghaus' review cycles to principled forgetting mechanisms and provides formal guarantees on stability, forgetting behaviour, and noise attenuation. The approach redefines core SSM parameters to enable time-aware forgetting and input-conditioned reviewing, resulting in a technically meaningful framework that scales linearly with sequence length and opens doors for efficient implementation across downstream tasks. Experimentally, DyG-Mamba demonstrates superior performance across 12 benchmarks in link prediction and node classification, achieving state-of-the-art accuracy while being an order of magnitude faster and using a quarter of the GPU memory compared to Transformer baselines, with robustness to up to 60% noisy edges and comprehensive evaluation under multiple negative sampling strategies.

The main weaknesses identified include several key issues. First, reviewers found insufficient justification for the specific learnable function form for $\Delta$t in Equation 11. They raised concerns about sensitivity to this choice and noted the lack of detailed ablation studies. The paper also failed to connect theoretical bounds with empirical evaluation.

Reviewers also noted that experiments were limited to modest-scale graphs. This raised questions about GPU efficiency on million-edge streams or with heterogeneous edge types. Critics viewed the approach as primarily applying Mamba architecture to DyG-Former's problem domain. They argued it lacked significant novel highlights beyond inherent Mamba advantages.

Additional concerns included the model's inability to distinguish when long history is unnecessary versus beneficial. The content-agnostic decay could potentially disregard temporally distant important signals. Reviewers also criticized the overemphasis on metaphorical explanations that felt stretched in technical contexts.

In response, the authors provided ablation studies demonstrating superior performance of their $\Delta$t function choice over alternatives, confirmed Lipschitz bound tightness through additional experiments, and addressed content-aware gating concerns by highlighting input-dependent parameterisation that functions similarly to Transformer Query-Key mechanisms. They also presented large-scale experiments on heterogeneous networks showing best MRR performance, refined theoretical descriptions to reduce metaphorical overemphasis, and expanded comparisons with contemporaneous works, while supplementary small-scale experiments demonstrated adaptability beyond long sequences.

Among the majority of reviewers their was support for acceptance and I endorse this view.